# SENSITIVITY-ADAPTIVE AUGMENTATION FOR ROBUST SEGMENTATION

## ABSTRACT

Achieving robustness in image segmentation models is challenging due to the fine-grained nature of pixel-level classification. These models, which are crucial for many real-time perception applications, particularly struggle when faced with natural corruptions. While sensitivity analysis can help us understand how input variables influence model outputs, applying it to natural and uncontrollable corruptions in training data is difficult. In this work, we present an efficient, sensitivity-based augmentation method to enhance robustness against natural corruptions. Our sensitivity analysis approach runs up to $10\times$ faster and requires up to $200\times$ less storage than previous approaches, enabling practical, on-the-fly estimation during training for a model-free augmentation policy. With minimal fine-tuning, our sensitivity-based augmentation method achieves improved robustness on both real-world and synthetic datasets compared to state-of-the-art data augmentation techniques in image segmentation tasks.

## 1 INTRODUCTION

Segmentation models are crucial in many applications, but they often face unpredictable and uncontrollable natural variations that can degrade their performance. For instance, mobile applications using segmentation for image reconstruction may encounter diverse noises due to varying environmental lighting, camera quality, and user handling. Similarly, autonomous vehicles and outdoor robots operate under a wide range of adverse weather conditions that are difficult to simulate accurately. Even in medical imaging, where conditions are more controlled, factors such as slight movements can introduce blur, affecting segmentation results. While poor-quality examples can sometimes be discarded and re-captured, such solutions are costly or impractical, especially in large-scale, ubiquitous use cases, with limited resources, and during real-time inference (e.g., failure in a navigating robot). Addressing these natural corruptions is challenging because they are hard to predict, simulate, or parameterize, yet they significantly impact model performance.

One common approach to enhance robustness against such corruptions is data augmentation, which artificially increases the diversity of training data by applying transformations to existing samples. While data augmentation is convenient and resource-efficient, its effectiveness depends on selecting the most beneficial augmentations. Ideally, we would know which augmentations a model is most sensitive to and focus on those to improve performance—in other words, sensitivity analysis. However, traditional sensitivity analysis methods are computationally expensive and resource-intensive (Shen et al., 2021), as shown in Table 1, making them impractical for large-scale or real-time applications. Existing methods like AutoAugment (Cubuk et al., 2019) and DeepAutoAugment attempt to optimize augmentation policies by training separate models, which adds significant overhead. Other state-of-the-art techniques rely on random augmentations (Cubuk et al., 2020; Muller & Hutter, 2021; Hendrycks et al., 2020), which are scalable but may not target the most impactful transformations for a given model.

In this paper, we propose a scalable, sensitivity-based augmentation approach for robustifying segmentation models against natural corruptions, including those not explicitly involved during training. Our approach performs a lightweight, online sensitivity analysis during training to identify the geometric and photometric perturbations, shown to be effective as "basis perturbations" (Shen et al., 2021), to which the model is most sensitive. In contrast to Shen et al. (2021), our sensitivity analysis is adaptive and significantly less resource intensive, allowing for practical implementation

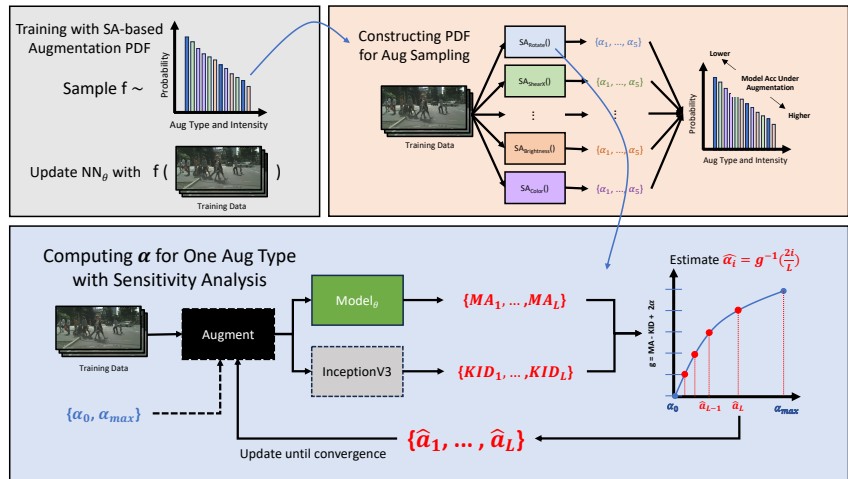

Figure 1: **Overview of our method.** We conduct sensitivity analysis using our Fast Sensitivity Analysis algorithm after a warmup period on clean data, then solve for $L$ discrete perturbation levels per perturbation type which the model is sensitive to. Finally, we augment training by sampling from the computed perturbation levels. Sampling weights are determined based off model performance on sensitive levels, where worse-performing levels are given higher probability of being sampled.

without the need for offline models or extensive computation. Figure 1 shows a high-level overview of our augmentation pipeline. Our method bridges the gap between the efficiency of random augmentation techniques and the effectiveness of policy-based augmentations guided by sensitivity analysis. Despite our focus on segmentation, our approach is general and can be applied to other tasks, architectures, or domains without significant modifications.

In experiments, we achieve up to a *6.20%* relative mIoU improvement in snowy weather and up to a *3.85%* relative mIoU improvement in rainy weather compared to the next-best method in zero-shot adverse weather evaluation on state-of-the-art architectures. We also show improvements on synthetic benchmarks and increased data efficiency compared to other augmentation methods as the size of the training set changes.

Our contributions are summarized as follows:

1. An efficient *adaptive* sensitivity analysis method for *online model evaluation* that iteratively approximates model sensitivity curves for speedup;
2. A comprehensive framework that leverages sensitivity analysis results to systematically improve the robustness of learning-based segmentation models;
3. Evaluation and analysis of our method on *unseen* synthetically perturbed samples, *naturally corrupted* samples, and ablated contributing factors to robustification.

## 2 RELATED WORKS

**Robustification Against Natural Corruptions**. The effect of natural corruptions on deep learning tasks is a well-explored problem, especially in image classification. Currently, image classification has a robust suite of benchmarks, including evaluation on both synthetic and natural corruptions (Hendrycks et al., 2020; Yi et al., 2021; Dong et al., 2020). Many works study correlations between image corruptions and various factors (Mintun et al., 2021; Hendrycks & Gimpel, 2017). Additionally, a popular approach to increasing robustness in the general case is through targeted adversarial training (Xiaogang Xu & Jia, 2021; Shu et al., 2021). Several approaches target model architecture (Schneider et al., 2020; Saikia et al., 2021; Myronenko & Hatamizadeh, 2020). Other approaches achieve robustness to natural corruptions via the data pipeline. Data augmentations are a popular method for increasing out-of-distribution robustness and many have now become standard practice (Geirhos et al., 2019; Rusak et al., 2020). Hendrycks et al. highlight that existing methods for generalization may not be consistently effective, emphasizing the need for robustness through addressing multiple distribution shifts (Hendrycks et al., 2021). In our work, we focus on studying and improving robustness in the context of semantic segmentation models due to natural corruptions

| Method | SA Time | Data Gen Time | Storage |
|--------|---------|---------------|---------|
| AdvSteer | 90.0±15.5 min | ∼ 48 hours | 2.4 TB |
| Ours | 9.6±0.2 min | - | 12 GB |

Table 1: **Runtime and Storage Comparison on Sensitivity Analysis of Shen et al. (2021), Compared to Ours.** Our approach enables the practical use of sensitivity analysis in online training as an augmentation policy. We compute each mean and standard deviation value in "SA Time" with 4 runs. Each sensitivity analysis iteration computes curves for 24 different augmentations at 5 levels each, for a total of 120 evaluation passes. Ours runs about $10\times$ faster and takes $200\times$ less storage.

using insights from previous work. Among findings from other works, we distinguish that our work focuses on improving natural corruption robustness in a segmentation, a common task with unique properties.

**Data Augmentation Techniques**. Data augmentation methods generate variants of the original training data to improve model generalization capabilities. These variants do not change the inherent semantic meaning of the image, and transformed images are typically still recognizable by humans. Within data augmentation methods, CutMix and AugMix widely-used augmentation techniques that augment by mixing variants of the same image (Hendrycks et al., 2020; Yun et al., 2019). Conversely, Franchi et al. (2021) introduces segmentation-specific augmentation approaches which utilize superpixels, or clusters of similar pixels, to maintain semantic object information. Other data augmentation methods have utilized augmentation policies based on neural networks to select productive augmentations (Olsson et al., 2021; Cubuk et al., 2019; Zheng et al., 2022), while other works have explored data augmentation for domain-specific tasks (Zhao et al., 2019; Zhang et al., 2023). For example, Zhao et al. (2019) explores learned data augmentation for biomedical segmentation tasks via labeling of synthesized samples with a single brain atlas. Zhang et al. (2023) explores data augmentation in specifically brain segmentation via combining multiple brain scan samples, similarly to Augmix and Cutmix. However, this work is reliant on additional annotations to augment regions of interest. In our work, we present a generalizable augmentation technique and show that performance boosts generalize well out-of-the-box on several domains.

## 3 METHODOLOGY

In general, sensitivity analysis examines how small fluctuations in the inputs affects the outputs of a system. In our augmentation approach, *the key idea is that sensitivity analysis can be used to sample augmentations uniformly with respect to impact on model performance, as opposed to sampling uniformly across the parameterized augmentation space*.

To quantify this for a given deep learning model, we need a metric for model performance and a metric for image degradation which is consistent across augmentation types. Choosing a model performance metric is straightforward; any bounded measure of accuracy (MA) where higher values are better suffices. As for the image degradation metric, we use Kernel Inception Distance (KID), introduced by Bińkowski et al. (2018) to reduce bias towards sample size. At a high level, we use KID to measure the "distance" between an original dataset and its perturbed version. KID does so by passing both datasets through a generalized Inception model, and computing the square Maximum Mean Discrepancy (MMD) between their respective features. The reduced sample size bias of KID allows us to approximate the image degradation metric without iterating through the full validation set.

By sampling augmentations to which the model is sensitive, we can improve robustness productively. We define the sensitivity of the model to changes in augmentation intensity as the ratio of the change in model accuracy to the change in KID:

$$\text{sensitivity} = \frac{\Delta MA}{\Delta KID} \tag{1}$$

Our goal is to identify augmentation intensities that result in high sensitivity—that is, small changes in the augmentation (as measured by KID) lead to large changes in model performance (MA). This indicates that the model is particularly sensitive to those augmentations, and training on them could

improve robustness. To formalize this, we seek to find a set of increasing, nontrivial augmentation intensities $\alpha_1 < \alpha_2 < \ldots < \alpha_L$ that maximize sensitivity. We define the local changes in accuracy and KID between consecutive intensities as:

$$\Delta\widehat{MA}(\alpha_i, \alpha_{i-1}) = MA(\alpha_{i-1}) - MA(\alpha_i) \tag{2}$$

$$\Delta\widehat{KID}(\alpha_i, \alpha_{i-1}) = \frac{D_{\text{KID}}(x_{\alpha_i}\|x_{\text{clean}}) - D_{\text{KID}}(x_{\alpha_{i-1}}\|x_{\text{clean}})}{D_{\text{KID}}(x_{\alpha_{\max}}\|x_{\text{clean}})} \tag{3}$$

Here, $MA(\alpha)$ is the model accuracy at augmentation intensity $\alpha$, and $D_{\text{KID}}(x_\alpha\|x_{\text{clean}})$ is the KID between the augmented data at intensity $\alpha$ and the original clean data. The normalization in $\Delta\widehat{KID}$ ensures that KID values are comparable across different augmentation types.

We then formulate an objective function $Q$ to find the set of intensities that maximizes sensitivity while ensuring adequate spacing between them:

$$Q = \arg\max_{\alpha_1,\ldots,\alpha_L} \min_{2\leq i\leq L} \left[\Delta\widehat{MA}(\alpha_i, \alpha_{i-1}) - \Delta\widehat{KID}(\alpha_i, \alpha_{i-1}) + \lambda(\alpha_i - \alpha_{i-1})\right] \tag{4}$$

In this equation, the term $\Delta\widehat{MA}(\alpha_i, \alpha_{i-1})$ represents the decrease in model accuracy between intensities $\alpha_{i-1}$ and $\alpha_i$. We subtract $\Delta\widehat{KID}(\alpha_i, \alpha_{i-1})$ to favor intensity intervals where accuracy drops more than the image degradation increases, thus indicating higher sensitivity. Furthermore, the regularization term $\lambda(\alpha_i - \alpha_{i-1})$ (with $\lambda > 0$) encourages spacing between intensities, preventing them from being too close together. In our implementation, $\lambda = 2$.

Our objective seeks to maximize the minimum value of this expression across all intervals, ensuring that even the least favorable interval is optimized.

To simplify the optimization, we introduce a function $g(\alpha)$:

$$g(\alpha) = 1 - MA(\alpha) - \frac{D_{\text{KID}}(x_\alpha\|x_{\text{clean}})}{D_{\text{KID}}(x_{\alpha_{\max}}\|x_{\text{clean}})} + \lambda\alpha \tag{5}$$

The set of $\alpha$ values which fulfills $Q$ has the following property: $g(\alpha_2) - g(\alpha_1) = g(\alpha_3) - g(\alpha_2) = \ldots = g(\alpha_L) - g(\alpha_{L-1})$; in other words, optimal $\alpha$ values are produced at equal intervals along the function $g$. Since $g(\alpha)$ is approximately monotonically increasing (as $MA(\alpha)$ decreases and $D_{\text{KID}}(x_\alpha, x_{\text{clean}})$ increases with increasing $\alpha$), and its values lie within a known range, we can approximate the solution as:

$$\alpha_i \approx g^{-1}\left(\frac{G_{\max} \cdot i}{L}\right), \quad i = 1, \ldots, L \tag{6}$$

where $G_{\max}$ is the maximum value of $g(\alpha)$ over the range of $\alpha$, and $g^{-1}$ is the inverse function. Since we choose $\lambda = 2$ in our implementation, $G_{\max} = 2$.

However, since we cannot explicitly compute $g^{-1}$ due to $g(\alpha)$ being unknown in closed form, we iteratively estimate the values of $\alpha_i$ using methods like the Piecewise Cubic Hermite Interpolating Polynomial (PCHIP), which is a spline estimation technique. By sampling a few initial points and fitting an interpolating function, we can estimate the intensities that satisfy our objective. We show the pseudocode for sensitivity analysis in Algorithm 2 of the appendix. Additionally, the iterative process for solving $\alpha$ values is visualized in Appendix Figure 12. Below, we show the full training routine involving Sensitivity Analysis in Algorithm 1.

**Resource differences from previous work in sensitivity analysis.** Previous sensitivity analysis methods (Shen et al., 2021) compute $g(\alpha)$ using a uniformly sampled set of $\alpha$ values across the entire augmentation space. This approach requires evaluating the model at many intensities and often necessitates offline generation of augmented datasets for each intensity and augmentation type. As a result, the storage complexity becomes the size of the original dataset multiplied by the number of

augmentation types and intensities, leading to substantial storage demands. In contrast, our method performs sensitivity analysis online during training and adaptively samples intensities based on the model's responses. By estimating $g(\alpha)$ iteratively and focusing only on necessary intensities, we eliminate the need for pre-generating augmented datasets. As a result, our approach only adds about 0.2 * (number of updates) * (evaluation time) amount of time to the total training pipeline, making the use of sensitivity analysis practical for on-the-fly augmentation policy during training.

---

**Algorithm 1:** Training with Sensitivity-Informed Augmentation.

---

**Data:** Training dataset $X_t$, Validation dataset $X_v$, Validation Rate $r_v$, SA Rate $r_{SA}$
**Result:** Trained semantic segmentation model

1  $N_V \leftarrow 0$ ;         // Number of validation rounds
2  $f(\cdot) \leftarrow Identity(\cdot)$ ;         // Augmentation transformation
3  Initialize network weights $\theta$;
4  **for** $i \leftarrow 1...max\_iter$ ;         // Training loop
5  **do**
6     $x_{ti} \leftarrow DataLoader(X_t)$;
7     **if** $p_f$ *is initialized* **then**
8         $f \sim p_f$ ;         // Sample aug PDF
9     **end**
10    $x_{ti}^{aug} \leftarrow f(x_{ti})$;
11    **if** $i \% r_v == 0$ **then**
12       **if** $i \% r_{SA} == 0$ ;         // Update Sensitivity Analysis
13       **then**
14          levels $\leftarrow []$ ;         // Store all $\alpha$ values
15          metrics $\leftarrow []$ ;         // Store all metrics
16          **for** *each augmentation type $f$* **do**
17              $\alpha_f, acc_f \leftarrow$ SensitivityAnalysis(f, $\theta$);   // Appendix: Algorithm 2
18              levels.append($\alpha_f$);
19              metrics.append($acc_f$);
20          **end**
21          levels = levels.sort() ;     // Sort based on descending metrics
22          $p_f \leftarrow$ BetaBinom(idx($f$), 0.75, 1.0) ;     // Categorical PDF by Acc
23       **end**
24       **for** $x_{vi} \leftarrow DataLoader(X_v)$ ;         // Validation loop
25       **do**
26          Compute clean validation metrics;
27       **end**
28    **end**
29  **end**

---

## 4 EXPERIMENTS

**Hardware.** Each experiment is conducted on four NVIDIA RTX A4000 GPUs and 16 AMD Epyc 16-core processors. Sensitivity analysis experiments are conducted on one GPU and 4 processors.

**Experiment Setup.** We use three different architectures across experimental results. For evaluation on real-world corruptions and data effiency, we train all experiments with the Segformer (Xie et al., 2021) backbone, a robust and state-of-the-art architecture for segmentation. For results on other architectures, a direct comparison of performance between PSPNet and Segformer architectures can be found in Section D.5 of the Appendix. Finally, for results in downstream fine-tuning from foundation model DinoV2 (Oquab et al., 2024), we use the original ViT (Dosovitskiy et al., 2021) architecture as the backbone. All methods are trained for 160k iterations regardless of approach, and only the best-performing checkpoints by mIoU (mean Intersection-over-Union by class) are used for evaluation in results. Additionally, nearly all models share the same set of augmentations, with the exception of IDBH (Li & Spratling, 2023), which uses an additional two augmentations (RandomFlip and RandomErase). We use official implementations for each method, and fix the random seed for each experiment such that they are reproducible. Full experiment configurations will be released

| Method | Weather // ACDC | | | Domain // IDD | | |
|---|---|---|---|---|---|---|
| | aAcc↑ | mIoU↑ | mAcc↑ | aAcc↑ | mIoU↑ | mAcc↑ |
| Baseline | 76.31 | 35.48 | 47.36 | 85.82 | 38.44 | 59.14 |
| AugMix | 79.57 | 40.90 | 52.74 | **86.52** | 40.50 | 62.43 |
| AutoAugment | 70.29 | 39.31 | 54.18 | 85.79 | **40.74** | 62.24 |
| RandAug | 78.46 | 39.07 | 52.32 | 85.54 | 38.99 | 59.82 |
| TrivialAug | 75.50 | 38.56 | 53.62 | 85.23 | 39.61 | 61.04 |
| IDBH | 78.65 | 41.67 | 53.65 | 86.49 | 40.48 | 61.74 |
| Ours | **80.16** | **45.45** | **57.58** | 85.76 | 40.33 | **63.03** |

Table 2: **Evaluation results on Unseen Real World Driving Datasets.** We conduct zero-shot evaluation of Cityscape models on both ACDC (Sakaridis et al., 2021) and IDD (Varma et al., 2019) datasets, which represent adverse weather and domain transfer to India respectively. Our method achieves clear improvements compared to other methods which require chained, more computationally expensive augmentations or external augmentation models in terms of generalization to real world scenarios, with relative mIoU improvement up to *9.07% on ACDC compared to the next-best, IDBH.*

alongside the code implementation for full reproducibility of results. More hyperparameter details for experiments can be found in Appendix Section C.

**Metrics.** We use three different metrics for evaluating the performance of a segmentation model: absolute pixel accuracy (aAcc), mean pixel accuracy (mAcc), and mean Intersection-over-Union (mIoU). Mean values are taken over object classes—thus, aAcc will be more susceptible to class imbalances, although it is the most intuitive.

| Method | Fog | | Rain | | Night | | Snow | |
|---|---|---|---|---|---|---|---|---|
| | aAcc↑ | mIoU↑ | aAcc↑ | mIoU↑ | aAcc↑ | mIoU↑ | aAcc↑ | mIoU↑ |
| Baseline | 89.70 | 55.10 | 87.41 | 42.82 | 54.39 | 14.89 | 83.23 | 41.22 |
| AugMix | 89.76 | 57.79 | **89.28** | 47.53 | 56.64 | 17.35 | 83.34 | 43.94 |
| AutoAugment | 77.06 | 56.18 | 75.52 | 42.66 | 57.14 | 20.65 | 71.83 | 40.94 |
| RandAug | 88.24 | 53.99 | 86.92 | 43.10 | 56.03 | 18.08 | 83.35 | 41.86 |
| TrivialAug | 85.79 | 55.16 | 84.35 | 41.26 | 54.52 | 17.02 | 77.99 | 42.64 |
| IDBH | 89.79 | 60.79 | 86.93 | 45.64 | 54.76 | 18.41 | **83.88** | 45.35 |
| Ours | **90.20** | **62.50** | 88.87 | **49.36** | **58.85** | **20.72** | 83.39 | **48.16** |

Table 3: **Evaluation of zero-shot adverse weather performance across data augmentation techniques.** We evaluate each data augmentation method across four different weather scenarios from the Adverse Conditions Dataset with Correspondences (ACDC) (Sakaridis et al., 2021) dataset. Each model is trained only with clean Cityscapes data with the Segformer (Xie et al., 2021) backbone. Our method, highlighted in grey, maintains the best performance across nearly all metrics for three out of four scenarios, with relative mIoU improvement over the next best method of up to *2.81% on fog, 3.85% on rain, and 6.20% on snow.*

### 4.1 EVALUATION ON REAL-WORLD CORRUPTIONS

To evaluate the robustness of our model in visual and graphics applications, we test on real-world adverse samples. While real-world adverse samples in most datasets are difficult to obtain, there are numerous real-world datasets for driving representing different cities and adverse weather scenarios.

We evaluate Cityscapes models with Segformer backbone on two real-world datasets: the Adverse Conditions Dataset with Correspondences (ACDC) (Sakaridis et al., 2021) dataset which represents adverse weather, and the India Driving Dataset (IDD) (Varma et al., 2019) which represents an alternative, more heterogeneous domain. IDD represents an alternative, but similar, domain in which visual appearances of vehicles, traffic, and scenery may slightly change, in addition to co-occurrences of classes. We emphasize that, for this experiment, models are only trained on Cityscapes, and evaluation on such scenarios can be interpreted as zero-shot generalization.

Overall performance on both ACDC and IDD datasets across multiple methods can be found in Table 2. We compare our results to six methods: a baseline model where no augmentation is performed, AugMix (Hendrycks et al., 2020), AutoAugment (Cubuk et al., 2019), RandAugment (Cubuk et al., 2020), and TrivialAugment (Muller & Hutter, 2021), and IDBH (Li & Spratling, 2023). On real-world

dataset evaluation for unseen weather and domain gap scenarios, our method shows improvements over the next best performing model across almost all metrics. We include a qualitative visualization of our model versus several other methods in Figure 6 of the appendix, which shows inference on a rainy weather sample. Amongst all methods, a common failure mode is the presence of windshield wipers in rainy weather. A visualization of this can be found in Appendix Section D.2.

A break-down the performance on the ACDC dataset by weather type in Table 3. In total, the ACDC dataset has four different weather scenarios: Fog, Rain, Night, and Snow, where the largest relative boost over next-best method, IDBH Li & Spratling (2023), (6.20%) is in Snow scenarios. In three out of four weather categories, our method outperforms other methods, with the exception of Night scenarios. AugMix achieves higher aAcc but lower mIoU than our method on Rain scenarios possibly due to class imbalances, such as the large number of pixels classified as "sky". While the total # of correct pixels is higher on AugMix, our method outperforms when averaged by class, on mIoU. Night scenario visibility corruption stems from lack of lighting, as opposed to the other three, which may have more differences in object appearances and blurring effects. While our method does not perform worse in mIOU, we do perform worse in aACC. This may suggest that the failure mode of our method in Night scenarios are due to smaller objects covering less pixel space.

**Special case: co-occurence of windshield wipers and rainy weather.** In the ACDC dataset, the rainy scenario evaluation set contains co-occurences with windshield-wiper occlusion. This case is interesting in that occlusions are not included in any experiments except those of IDBH. In qualitative results, we observe that our method handles windshield wiper occlusions just as well, if not better, than IDBH. In Figure 2, we show an example of this, where our method shows comparatively less artifacts in the building and sky, despite not having been trained on occlusion (RandomErase) augmentations.

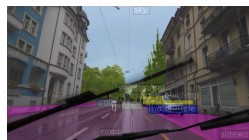 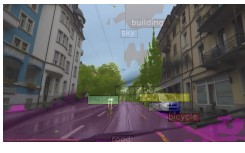 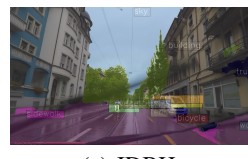 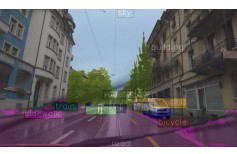

(a) Ground Truth.      (b) AutoAugment.      (c) IDBH.      (d) Ours.

Figure 2: **Special case on ACDC prediction: windshield wiper occlusion.** We observe a special case of natural corruptions in rainy weather which cannot be directly simulated by the existing set of perturbations: physical occlusion by windshield wipers. While IDBH involves random occlusion during training, ours does not.

### 4.2 EVALUATION ON DATASETS

The results in previous experiments show the efficacy of our method in context of driving domains. In this experiment, we demonstrate that our method also shows improvements across several datasets and visual computing domains compared to SOTA.

We evaluate our method on six semantic segmentation datasets: ADE20K (Zhou et al., 2019), VOC2012 (Everingham et al., 2012), POTSDAM (for Photogammetry & Sensing), Cityscapes (Cordts et al., 2016), Synapse (Landman et al., 2015), and A2I2Haze (Narayanan et al., 2023). POTSDAM is a remote sensing datasets taken from aerial views, with classes focusing on classification of buildings, roads, trees, etc. POTSDAM describes aerial imagery in Potsdam, Germany. Cityscapes is a popular benchmark dataset for segmentation in urban traffic scenes, with annotations describing classes such as terrain, human, and vehicle types. ADE20K and VOC2012 are generic datasets describing everyday life and objects, with both indoor and outdoor scenes. Synapse is a medical imaging dataset of clinically-acquired CT scans. In our experiments, we use abdomen data and classify organs. A2I2Haze is a dataset representing outdoor clear and hazy data collected from unmanned robots for scene understanding. We use the UGV, or Unmanned Ground Vehicle data in our experiments, which is similar to autonomous driving datasets except in more heterogeneous outdoor environments.

In Table 4, we show mIoU performance of our method versus the next-best augmentation technique, the SOTA baseline. We evaluate on clean data and three different synthetic scenarios: individual transformations from the basis augmentations at uniform param-

| Dataset | Type | Method | Clean | | Basis Aug | | AdvSteer | | IN-C | |
|---|---|---|---|---|---|---|---|---|---|---|
| | | | aAcc↑ | mIoU↑ | aAcc↑ | mIoU↑ | aAcc↑ | mIoU↑ | aAcc↑ | mIoU↑ |
| ADE20K | General | TrivialAug | 75.420 | 32.580 | 69.559 | 27.083 | 41.783 | 9.188 | 61.495 | 18.668 |
| | | IDBH | **76.220** | **33.950** | 72.752 | 30.651 | 40.557 | 9.475 | **61.971** | **19.091** |
| | | Ours | 76.110 | 33.790 | **74.285** | **31.922** | **43.075** | **9.628** | 61.280 | 18.721 |
| VOC2012 | General | TrivialAug | 90.090 | 57.900 | 87.837 | 52.340 | 75.350 | 20.338 | **82.884** | 36.080 |
| | | IDBH | 90.610 | 60.570 | 89.262 | 56.876 | 69.843 | 20.810 | 81.819 | 36.933 |
| | | Ours | **90.800** | **61.140** | **89.555** | **58.183** | 69.690 | **21.470** | 82.519 | **38.834** |
| POTSDAM | Aerial | TrivialAug | 84.360 | 67.820 | 77.649 | 55.763 | **55.817** | **34.282** | **55.866** | **36.967** |
| | | IDBH | 84.280 | **68.690** | 79.392 | 63.757 | 22.675 | 14.975 | 46.413 | 30.123 |
| | | Ours | **84.550** | 68.450 | **82.590** | **66.065** | 44.817 | 29.983 | 54.275 | 36.416 |
| A2I2Haze | UGV | TrivialAug | 98.730 | 69.180 | 97.317 | 51.800 | 85.598 | **22.225** | 97.363 | 46.502 |
| | | IDBH | 98.680 | 69.300 | 98.346 | 64.615 | 85.545 | 19.490 | 97.368 | 45.970 |
| | | Ours | **98.790** | **70.290** | **98.613** | **67.919** | **89.482** | 21.843 | **97.407** | **49.805** |
| Cityscapes | Driving | TrivialAug | 95.570 | 74.300 | 86.117 | 56.952 | 69.785 | **30.593** | 82.664 | 44.332 |
| | | IDBH | 95.530 | 73.930 | 93.160 | 68.052 | **71.932** | 29.388 | **83.041** | 44.225 |
| | | Ours | **95.780** | **75.530** | **94.305** | **71.539** | 68.468 | 28.070 | 82.435 | **45.066** |
| Synapse | Medical | TrivialAug | 98.890 | 62.000 | 97.939 | 49.237 | **97.243** | **32.182** | 98.425 | 51.512 |
| | | IDBH | 99.150 | 67.720 | 98.912 | 63.504 | 95.143 | 29.760 | **98.486** | 53.475 |
| | | Ours | **99.250** | **71.380** | **99.082** | **68.828** | 90.282 | 30.310 | 96.779 | **56.013** |

Table 4: **Performance evaluation of our method vs. SOTA on synthetic scenarios across 6 different datasets.** We evaluate our method and SOTA on ADE20K (Zhou et al., 2019), VOC2012 (Everingham et al., 2012), POTSDAM (for Photogammetry & Sensing), A2I2Haze (Narayanan et al., 2023), Cityscapes (Cordts et al., 2016), and Synapse (Landman et al., 2015) datasets, across three synthetic corruption scenarios: individual basis augmentations (Basis Aug), compositions of photometric augmentations produced by sensitivity analysis in Adversarial Steering (AdvSteer) (Shen et al., 2021), and the synthetic augmentation benchmark ImageNet-C (IN-C) (Hendrycks & Dietterich, 2019). Our method consistently achieves improved performance on synthetic corruption benchmarks while still maintaining or even improving clean evaluation accuracy.

eter intervals (Basis Aug), the combined perturbation benchmark from Shen et al. (2021) (AdvSteer), and ImageNet-C (IN-C) (Hendrycks & Dietterich, 2019) corruptions. *On the synthetic benchmark ImageNet-C (Hendrycks & Dietterich, 2019), our model achieves improved scores, particularly in the robotics and medical domains.* Our method performed worse primarily in the AdvSteer benchmark of Table 4, notably for Cityscapes and Synapse. This may be due to the sheer intensity of benchmark corruption—the AdvSteer benchmark applies a combination of intense perturbations (not the same as the augmentations used during training), resulting in an extreme case from the original distribution. This may be related to degraded performance on Night scenarios in ACDC evaluation, as both scenarios heavily corrupt visibility based on color. Examples of the AdvSteer benchmark corruptions can be found in Appendix Section D.4.

Qualitative results on Synapse with synthetic motion blur between our method and next best, TrivialAugment, can be observed in Figure D. We emphasize that our method is not necessarily bound to image segmentation—we find similar boosts in performance in classification (see Appendix Section 9).

| | ViT+DinoV2 | | |
|---|---|---|---|
| Method | aAcc↑ | mAcc↑ | mIoU↑ |
| Baseline | 77.65 | 45.83 | 32.70 |
| Augmix | 79.99 | 51.63 | 41.38 |
| AutoAugment | 81.18 | 55.93 | 43.65 |
| RandAugment | 80.42 | 54.02 | 43.25 |
| TrivialAugment | 82.56 | 54.27 | 43.58 |
| IDBH | **84.45** | 60.22 | 48.69 |
| Ours | 84.13 | **62.92** | **49.82** |

Table 5: **Performance of Cityscapes models on *unseen* ACDC weather evaluation set across different augmentation methods, when fine-tuned from DinoV2** (Oquab et al., 2024) with ViT (Dosovitskiy et al., 2021) backbone.

### 4.3 DOWNSTREAM FINETUNING WITH FOUNDATION MODELS

A popular choice for boosting feature robustness is fine-tuning downstream tasks from foundation models. In these experiments, we examine how our approach can complement robustness provided by foundation models when fine-tuning on downstream tasks. We first initialize a distilled DinoV2 (Oquab et al., 2024) model on the ViT-Small (ViT-S) architecture, then fine-tune on the semantic segmentation task with Cityscapes. We choose Cityscapes due to the availability of real-world corrupted images (ACDC and IDD) to evaluate on. In our experiments, we observe an 2.32% mIoU improvement over the next best method, IDBH. While the largest boost in robustness stem from

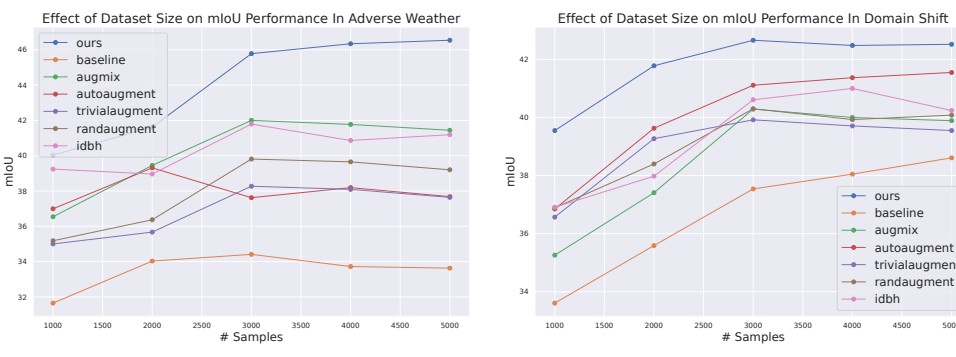

(a) # Samples vs. ACDC mIoU performance.  (b) # Samples vs. IDD mIoU performance.

Figure 3: **Comparison of Ours vs. SOTA Data Augmentation Methods**: Ours (top, blue) outperforms all others with performance improving as the number of samples increases, while other methods plateau on both (a) adverse weather data (ACDC) and domain shifted data (IDD).

robust foundation model features, our results suggest that our method can complement approaches centered around model architecture (such as Segformer).

## 4.4 DATA EFFICIENCY

We also analyze data efficiency of our method in comparison to other data augmentation methods by training various Segformer models with varying training dataset sizes. For each method in Table 2, we train five models with training dataset sizes of 1000, 2000, 3000, 4000, and 5000 samples from the Cityscapes dataset. We plot the progression of mIoU (Minimum Intersection over Union) performance (higher the better) on (a) adverse weather data (ACDC) and (b) the domain shift setting (IDD), as shown in Figure 3. Our method, in blue, shows consistent improvement on adverse weather and domain shift evaluation with increasing number of samples, and maintains best mIoU performance across each # of samples slice, suggesting that our method is more data efficient than others. Interestingly, not all methods show increased robustness to adverse weather as number of samples increases for training, indicating that in some cases, scaling data may not necessarily mean increased robustness.

## 4.5 ABLATION STUDY

We examine several variants of our method to determine the impact of individual components in an ablation study: a baseline trained only with random cropping, a variant of our method using only geometric augmentations, a variant of our method using only photometric augmentations, a variant of our method without clean training warmup, and a variant of our method using uniform sampling instead of the Beta-Binomial sampling described in Algorithm 1. Uniform sampling of augmentation parameters computed with sensitivity analysis decreases generalization to both synthetic and real-world corruption benchmarks by small margins. In addition, training without clean warmup produces similar results to that with warmup, suggesting that warmup is optional. In our case, warming up with clean evaluation reduces the total number of sensitivity analysis updates, making warm-up with clean evaluation marginally less resource expensive ( 0.5 GPU hours total). Interestingly, while clean performance remains largely the same across all models, the largest decrease in performance on unseen corruption benchmarks comes from the lack of photometric augmentations.

To examine generalization of photometric robustness over training, we plot the $g$ values computed from Equation 5 across training for our Cityscapes experiments in Figure 4. One curve is plotted per component for RGB, HSV, Noise, and Blur corruptions. Note that the components in this Figure are based on individual color channels and are separate from those used during training. From this visualization, we observe that Hue curves (teal, center) are most volatile during training, with most sensitive augmentation parameters falling towards $\alpha$ values close to 1.0 in the beginning of training. As the model generalizes, the Hue curve converges slowly towards $\alpha$ values centered around 0.5, similarly to other curves. *This suggests that Hue is a significant factor in model robustness*, whilst other channels are largely stagnant as models generalize over training.

| Method | Clean | | Basis Aug | | AdvSteer | | IN-C | | ACDC | |
|---|---|---|---|---|---|---|---|---|---|---|
| | aAcc↑ | mIoU↑ | aAcc↑ | mIoU↑ | aAcc↑ | mIoU↑ | aAcc↑ | mIoU↑ | aAcc↑ | mIoU↑ |
| Baseline | 95.610 | 75.130 | 92.042 | 65.319 | 62.040 | 21.995 | 79.437 | 38.362 | 78.49 | 37.54 |
| Ours$_{\sim g}$ | 95.780 | **75.500** | 93.405 | 68.877 | **71.070** | 27.997 | 83.032 | 44.385 | 78.13 | 43.69 |
| Ours$_{\sim p}$ | 95.740 | 75.210 | 92.544 | 69.002 | 64.907 | 22.437 | 80.817 | 40.876 | 75.74 | 37.97 |
| Ours$_{\sim Warmup}$ | **95.830** | 75.430 | **94.458** | **71.891** | 69.138 | 28.472 | 84.438 | 45.849 | 79.78 | 44.66 |
| Ours$_{\sim Uniform}$ | 95.740 | 75.200 | 94.304 | 71.213 | 69.678 | 27.235 | **85.135** | **46.219** | **80.95** | 43.17 |
| Ours | 95.790 | 75.100 | 94.439 | 71.665 | 70.605 | **28.895** | 83.844 | 45.617 | 80.13 | **44.67** |

Table 6: **Ablation study results** comparing different variants of our method. We compare: (1) a baseline trained with no augmentations, (2) a variant of our method that only augments with photometric augmentations (Ours$_{\sim g}$), (3) a variant of our method that only uses geometric augmentations (Ours$_{\sim p}$), (4) a variant of our method trained without clean training warmup, (5) a variant of our method with uniform augmentation (Ours$_{Uniform}$) of computed sensitivity analysis values $\alpha$, and (6) our full method combining informed probability sampling, and adaptive sensitivity analysis, and all augmentation types (Ours).

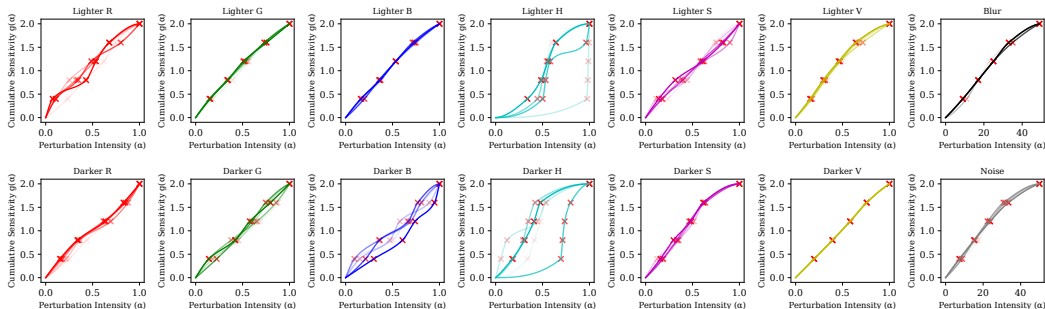

Figure 4: **Cumulative sensitivity curves (g values) throughout training of Cityscapes.** We visualize how the estimated cumulative sensitivity curve, Equation 5, changes for RGB, HSV, Gaussian blur, and Gaussian noise during augmented training. In this plot, the most recent curve is opaque, while others decrease in opacity in order of recency. The red X markers indicate the values at which $\alpha$ values are selected (horizontal axes). Surprisingly, most curves remain largely stagnant throughout training, with the exception of Hue in HSV (teal, center), which changes drastically as the model generalizes. This may suggest that Hue is a major factor in model generalization. Ablation study results in Table 6 support this, where the model trained without photometric augmentations demonstrate a significant decrease in performance.

## 5 DISCUSSION AND CONCLUSION

In this paper, we present a method for sensitivity-informed augmented training for semantic segmentation. Our method combines the information granularity of sensitivity analysis-based methods and the scalability of data augmentation methods, which run on-the-fly during training. In our results, we show that our method achieves improved robustness on zero-shot real-world adverse weather and domain shift scenarios, in addition to improvements on synthetic benchmarks like ImageNet-C. Additionally, evaluation on real world datasets show clear improvements over current SOTA methods for augmentation. Our model can complements other approaches for model robustness such as architecture design and downstream fine-tuning.

Currently, a limitation of our work is that our method does not address gaps in low-lighting scenarios. Future work can explore occlusion and low-lighting techniques for segmentation, as both cases resulted in degraded performance for all methods. Additionally, our method treats all augmentation types as equal, in that weighting of augmentation is uniform across types—sensitivity analysis is used to update the intensity values $\alpha$ only for online sampling. From our ablation study, we show that uniform sampling matters little in context of our method. However, future work dissecting whether all augmentations are equal, especially photometric augmentations, will be useful especially for unseen scenarios in robotics.

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
