# A   APPENDIX

# B   SENSITIVITY ANALYSIS PSEUDOCODE

---

**Algorithm 2:** Fast Sensitivity Analysis

---

**Data:** Number of levels $L$, Uncertainty threshold $\epsilon$
**Result:** Perturbation Levels $\{\alpha_1, ..., \alpha_{L-1}\}$

1  $g(\alpha) \leftarrow$ Equation 5;
2  points $\leftarrow \{(0,0), (\alpha_L, 2)\}$;
3  **loop**
4      $\hat{c} \leftarrow$ PCHIP(points);
5      **for** $i \leftarrow 1...L-1$ **do**
6          $\alpha_i \leftarrow$ Estimate($\hat{c}, 2i/L$);
7          $(y_l, y_u) \leftarrow$ Estimate upper and lower y-values of $\hat{c}$ at $x = \alpha_i$;
8          $\hat{c}_l \leftarrow$ PCHIP(points.insert($y_l$));
9          $\hat{c}_u \leftarrow$ PCHIP(points.insert($y_u$));
10         $\alpha_{i_l} \leftarrow$ Estimate($\hat{c}, y_l$);
11         $\alpha_{i_u} \leftarrow$ Estimate($\hat{c}, y_u$);
12         $\epsilon_i \leftarrow (\alpha_{i_u} - \alpha_{i_l})/2$;
13     **end**
14     $\alpha^*, \epsilon^* \leftarrow$ Choose level with max $\epsilon_i$;
15     **if** $\epsilon^* < \epsilon$ **then** Break loop;
16     points.insert($(\alpha^*, g^*(\alpha^*))$);
17 **end**;

---

## C   DETAILED EXPERIMENT HYPERPARAMETERS

| Method | Max Iters | LR | Optimizer | Augmentations | Batch Size | Backbone |
|---|---|---|---|---|---|---|
| Baseline | 160,000 | 6e-05 | AdamW | RandomCrop | 1 | SegFormer-b0 |
| Augmix | 160,000 | 6e-05 | AdamW | RandomCrop, Contrast, Equalize, Posterize, Rotate, Solarize, Shear X, Shear Y, Translate X, Translate Y, Color, Contrast, Brightness, Sharpness | 1 | SegFormer-b0 |
| AutoAugment | 160,000 | 6e-05 | AdamW | RandomCrop, Contrast, Equalize, Posterize, Rotate, Solarize, Shear X, Shear Y, Translate X, Translate Y, Color, Contrast, Brightness, Sharpness | 1 | SegFormer-b0 |
| RandAug | 160,000 | 6e-05 | AdamW | RandomCrop, Contrast, Equalize, Posterize, Rotate, Solarize, Shear X, Shear Y, Translate X, Translate Y, Color, Contrast, Brightness, Sharpness | 1 | SegFormer-b0 |
| TrivialAug | 160,000 | 6e-05 | AdamW | RandomCrop, Contrast, Equalize, Posterize, Rotate, Solarize, Shear X, Shear Y, Translate X, Translate Y, Color, Contrast, Brightness, Sharpness | 1 | SegFormer-b0 |
| IDBH | 160,000 | 6e-05 | AdamW | RandomCrop, Contrast, Equalize, Posterize, Rotate, Solarize, Shear X, Shear Y, Translate X, Translate Y, Color, Brightness, Sharpness, **RandomFlip**, **RandomErasing** | 1 | SegFormer-b0 |
| Ours; $r_v = 1600$; $r_{SA} = 9600$; $Warmup = 6400$ | 160,000 | 6e-05 | AdamW | RandomCrop, Contrast, Equalize, Posterize, Rotate, Solarize, Shear X, Shear Y, Translate X, Translate Y, Color, Contrast, Brightness, Sharpness | 1 | SegFormer-b0 |

Table 7: **Experiment hyperparameters for Table 2 and Table 4 .** All experiments are trained under similar hyperparameter settings, with each evaluation conducted on the *highest-performing mIoU checkpoint*. In comparisons, we prioritize official implementations released by authors and avoid re-implementations. Additionally, most comparisons use the same set of augmentations to ours, with the exception of IDBH Li & Spratling (2023), whose original implementation includes RandomFlip and RandomErasing. For all experiments, we use the SegFormer-b0 backbone Xie et al. (2021), which is a recent state-of-the-art segmentation-specialized architecture.

# D  QUALITATIVE RESULTS ON SYNAPSE

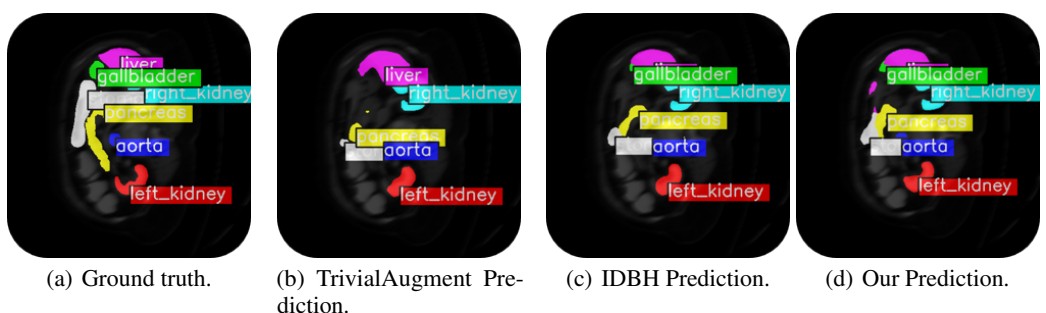

(a) Ground truth.  (b) TrivialAugment Prediction.  (c) IDBH Prediction.  (d) Our Prediction.

Figure 5: **Qualitative evaluation on multi-organ segmentation with motion blur corruption.** We show predictions on a motion-blurred sample from the Synapse (Landman et al., 2015) dataset for TrivialAugment (b), IDBH (c), and Our method (d), against the ground truth (a). Our method is able to segment right and left kidneys, liver, and aorta accurately. In contrast, the TrivialAugment prediction is unable to distinguish both kidneys.

## D.1  QUALITATIVE RESULTS ON RAINY DATA

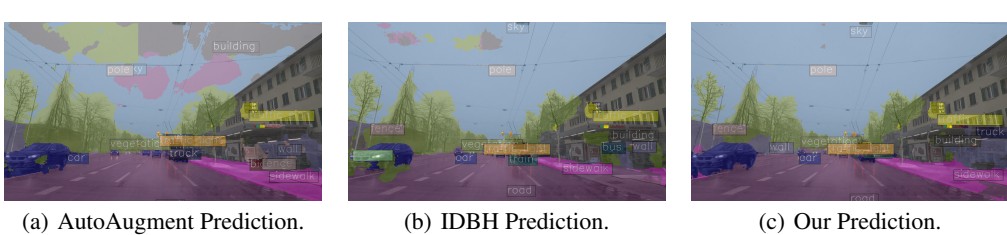

(a) AutoAugment Prediction.  (b) IDBH Prediction.  (c) Our Prediction.

Figure 6: **Qualitative comparison on snowy urban driving sample between AutoAugment Cubuk et al. (2020), IDBH Li & Spratling (2023), and Ours.** In this example, each method (AutoAugment, IDBH, Ours) is trained on clean Cityscapes data representing sunny weather, then evaluated on adverse weather samples. Despite not having rainy data in the training set, our method is able to segment the driving noticeably clearer than other methods. In particular, other methods consistently struggle to segment the vehicle confidently.

## D.2 SPECIAL CASE: WINDSHIELD WIPER OCCLUSION

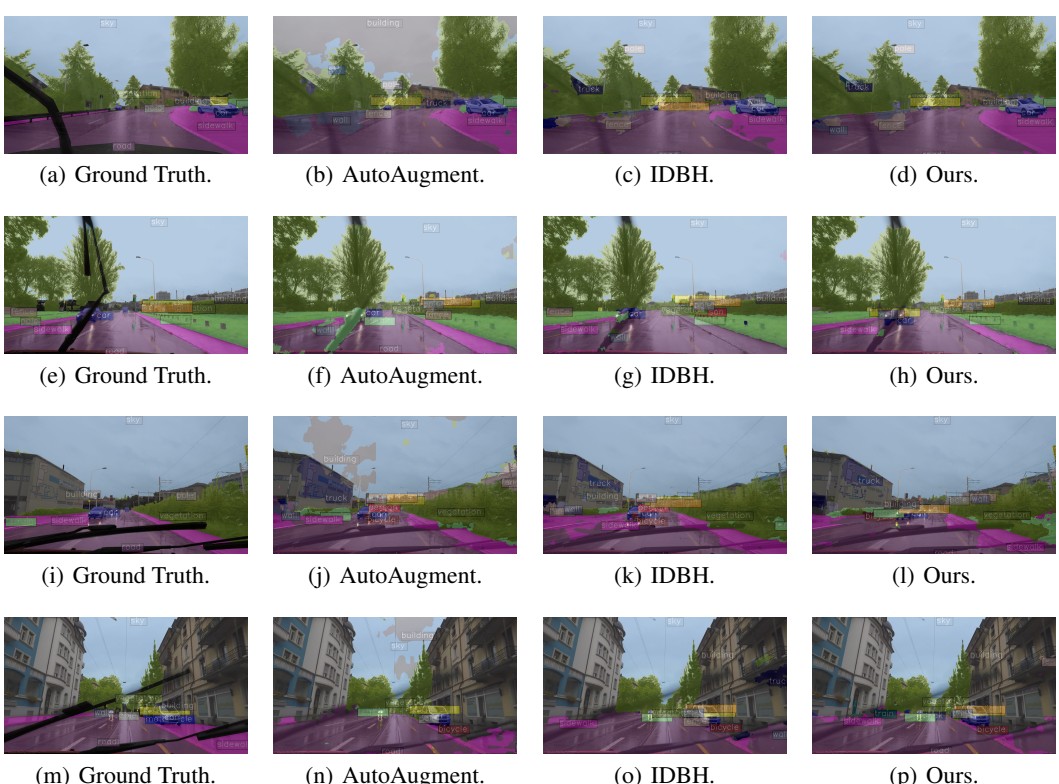

(a) Ground Truth.     (b) AutoAugment.     (c) IDBH.     (d) Ours.

(e) Ground Truth.     (f) AutoAugment.     (g) IDBH.     (h) Ours.

(i) Ground Truth.     (j) AutoAugment.     (k) IDBH.     (l) Ours.

(m) Ground Truth.     (n) AutoAugment.     (o) IDBH.     (p) Ours.

Figure 7: **More examples of special case on ACDC prediction: windshield wiper occlusion.**

## D.3 DETAILS ON BASIS AUGMENTATIONS

Previous work in robustification showed that learning with a set of "basis perturbations" (BP) significantly improved zero-shot evaluation against unseen corruptions Shen et al. (2021) for image classification and regression tasks, such as vehicle steering prediction. The intuition behind basis perturbations is that the composition of such perturbations spans a much larger space of perturbations than may be observed in natural corruptions; observed zero-shot performance boosts on unseen corruptions subsequently might be attributed to learning a model which is robust to basis perturbations. In our method, we extend this concept and introduce a more generalized and larger set of basis perturbations in sensitivity analysis to determine the most productive augmentation during training.

Let $D = \{Positive, Negative\}$ describe the set of augmentations applied in either a positive (lighter) direction or negative (darker) to either one channel of an image or a parameter of an affine transformation applied to an image.

Let $P = \{R, G, B, H, S, V\}$ describe the set of channels in RGB and HSV color spaces which may be perturbed; in other words, these augmentations are *photometric*.

Then, let $G = \{ShearX, ShearY, TranslateX, TranslateY, Rotate\}$ denote affine, or *geometric*, transformations which are parameterized by a magnitude value.

Finally, let $Z = \{Noise, Blur\}$ be the set of augmentations not applied along channel dimensions. Specifically, we use Gaussian Noise and Gaussian Blur.

Thus, the set of all basis augmentations $A_B$ used in robustification is $A_B = \{D \times P + G + Z\}$.

To compute lighter or darker channel augmentations of RGB or HSV channels, we use linear scaling. Let the range of a channel be $[v_{\min}, v_{\max}]$. For lighter channel augmentations, we transform the

channel values $v_C$ by an intensity factor $\alpha$ like so:

$$v'_C = \alpha v_{\max} + (1 - \alpha) \cdot v_C$$

Likewise, for darker channel augmentations, the transformation can be described like so:

$$v'_C = \alpha v_{\min} + (1 - \alpha) \cdot v_C$$

The default values are $v_{\min} = 0$ and $v_{\max} = 255$. For $H$ channel augmentations, we set the maximum channel values to be $180$. For $V$ channel augmentations, we set the minimum channel values to be $10$ to exclude completely dark images.

Affine transformations can be represented as a $3 \times 3$ matrix, which, when multiplied with a 2-dimensional image, produces a geometrically distorted version of that image. Affine transformation matrices are typically structured in the form:

$$M = \begin{bmatrix} 1 & Shear_X & T_x \\ Shear_Y & 1 & T_y \\ 0 & 0 & 1 \end{bmatrix}$$

for shear and translation transformations. For rotations where the center of the image is fixed as the origin point $(0, 0)$, the transformation matrix is defined as:

$$M_{rot} = \begin{bmatrix} cos\theta & -sin\theta & 0 \\ sin\theta & cos\theta & 0 \\ 0 & 0 & 1 \end{bmatrix}$$

To account for padded values in images after affine transformations, we zoom in images to the largest rectangle such that padded pixels are cropped out.

All augmentations are parameterized by a magnitude value ranging from 0 to 1. A magnitude value of 1 corresponds to the most severe augmentation value. More details on exact parameter value ranges can be found in the appendix. Conversely, a magnitude value of 0 produces no changes to the original image, and can be considered an identity function. We account for the symmetry of these augmentation transformations by considering both positive values and negative values as separate augmentations. The fast adaptive sensitivity analysis algorithm introduced in the next section relies on the property that increasing magnitude corresponds to increasing "distance" between images. Thus, augmentations cannot simply span the value ranges -1 to 1, and we separate them instead to different augmentations (positive and negative).

We apply these augmentations on-the-fly in online learning rather than generating samples offline. Doing so greatly reduces the offline storage requirement by one order of magnitude. Suppose $L$ intensity levels are sampled for each basis augmentation. Then, offline generation of perturbed data requires up to $L \times 2 \times (|P| + |G|) + 2 = 24L$ additional copies of the original clean dataset. *With online generation, we avoid offline dataset generation entirely* and only need the original clean dataset to be stored, similar to standard vanilla learning.

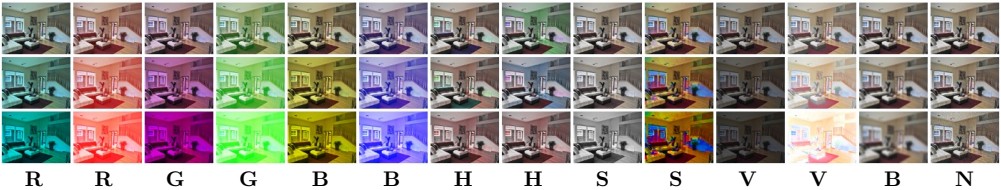

| R | R | G | G | B | B | H | H | S | S | V | V | B | N |

Figure 8: **Visualization of each photometric augmentation transformation** on a bedroom image. Up ↑ indicates the "lighter", positive direction and ↓ indicates the "darker", negative direction. "B" and "N" indicate blur and noise, respectively.

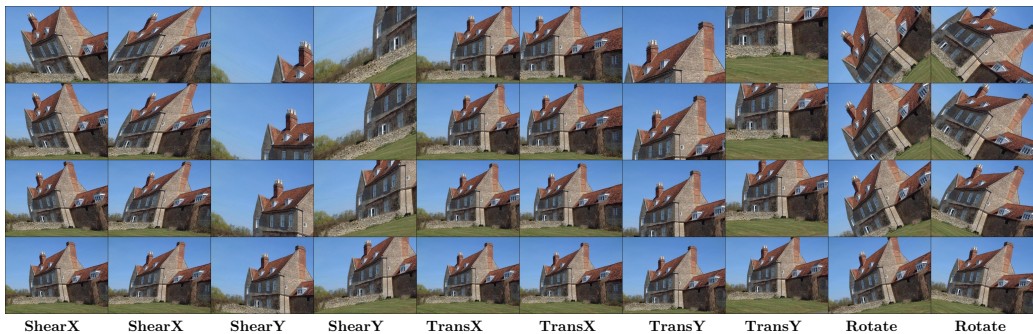

| ShearX | ShearX | ShearY | ShearY | TransX | TransX | TransY | TransY | Rotate | Rotate |

Figure 9: **Visualization of various geometric augmentations** applied to a sample image of a house. We use the following geometric transformations in our sensitivity analysis scheme, which are also analogous to the set of transformations used by other methods Cubuk et al. (2019); Zheng et al. (2022). Up arrows indicate augmentation in the *positive*, or left, direction, while down arrows indicate augmentation in the *negative*, or right, direction.

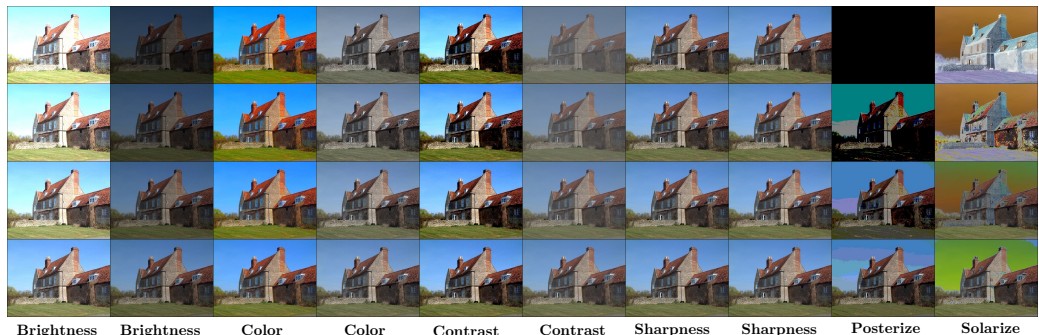

| Brightness | Brightness | Color | Color | Contrast | Contrast | Sharpness | Sharpness | Posterize | Solarize |

Figure 10: **Additional augmentation types used in sensitivity analysis, which are used in other methods such as AutoAugment.** While these photometric tranformations are used in other methods, the transformations also overlap with the photometric transformations shown in Figure 8, namely HSV perturbations. However, we still conduct sensitivity analysis evaluation on these transformations for completion.

### D.4 ADVSTEER BENCHMARK EXAMPLES

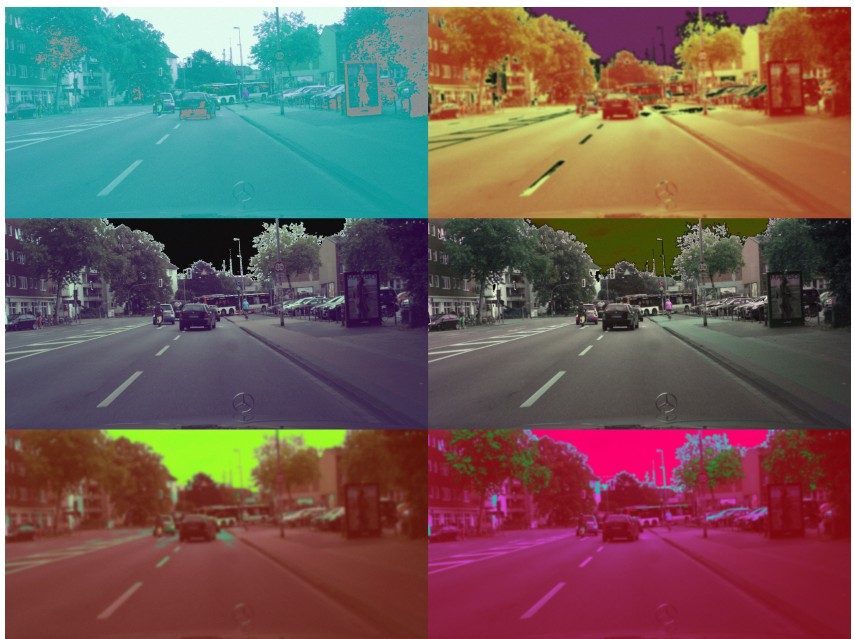

Figure 11: AdvSteer benchmark examples.

## D.5 CLEAN PERFORMANCE ON DIFFERENT BACKBONES

| | PSPNet Zhao et al. (2017) | | | SegFormer Xie et al. (2021) | | |
|---|---|---|---|---|---|---|
| Method | aAcc↑ | mAcc↑ | mIoU↑ | aAcc↑ | mAcc↑ | mIoU↑ |
| Baseline | 63.770 | 48.695 | 35.715 | 86.825 | 57.280 | 48.365 |
| Augmix | 94.770 | 74.400 | 66.740 | 95.520 | 81.430 | 73.390 |
| AutoAugment | **95.130** | 77.210 | 69.630 | 95.550 | 81.390 | 73.820 |
| RandAugment | 95.060 | 76.770 | 69.360 | 95.610 | 82.390 | 74.560 |
| TrivialAugment | 95.090 | 75.930 | 68.620 | 95.640 | 83.210 | 75.130 |
| Ours | 95.100 | **79.320** | **71.840** | **95.880** | **84.070** | **76.330** |

Table 8: **Comparison of clean evaluation performance across different augmentation methods on Cityscapes.** We evaluated our sensitivity-informed augmentation method against popular benchmarks on PSPNet and SegFormer. The baseline represents training with no augmentations.

## D.6 RESULTS ON CUB DATASET FOR CLASSIFICATION

| | InceptionV3 | | | |
|---|---|---|---|---|
| Method | Clean | Basis Aug | AdvSteer | IN-C |
| Baseline | 41.647 | 15.965 | 3.679 | 20.501 |
| Augmix | 35.865 | 15.274 | 4.810 | 20.394 |
| AutoAugment | 16.793 | 7.219 | 2.575 | 8.158 |
| TrivialAugment | 33.914 | 13.338 | 4.229 | 17.586 |
| RandAugment | 36.624 | 15.466 | 4.821 | 19.345 |
| Ours | **47.670** | **18.122** | **5.276** | **21.842** |

Table 9: Performance on CUB (Wah et al., 2011) dataset with InceptionV3 (Szegedy et al., 2016) backbone.

## D.7 FAST SENSITIVITY ANALYSIS ILLUSTRATION

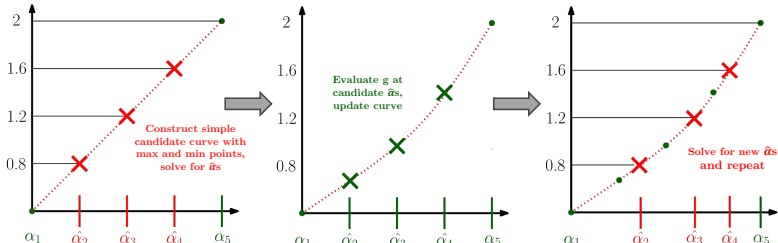

Figure 12: **Illustration of fast sensitivity analysis.** Each iteration of the fast sensitivity can be intuitively visualized. Since we can assume general monotonicity of the curve, we first initialize a candidate curve (a line in the first iteration). We solve for the candidate perturbation levels $\hat{\alpha}$ based on the solution in Equation 6. In the next step (middle), we evaluate the candidate level with the greatest uncertainty and adjust the candidate curve, the dotted red line, using PCHIP on the evaluated levels, which are guaranteed to be true points along the function $g$ from Equation 5. In the next step (right), we use the new curve and solve for new candidate levels, repeating the process in the previous two steps until the maximum uncertainty of any candidate level values falls below a threshold of 0.05.

## D.8  SENSITIVITY ANALYSIS COMPUTED CURVE COMPARISON

| Perturb | Method | $p_1$ | $p_2$ | $p_3$ | $p_4$ |
|---|---|---|---|---|---|
| $R_\uparrow$ | Baseline | 0.100 | 0.300 | 0.500 | 0.700 |
| | Adaptive | 0.149 | 0.253 | 0.399 | 0.604 |
| $G_\uparrow$ | Baseline | 0.100 | 0.200 | 0.400 | 0.600 |
| | Adaptive | 0.103 | 0.204 | 0.395 | 0.619 |
| $B_\uparrow$ | Baseline | 0.200 | 0.300 | 0.500 | 0.700 |
| | Adaptive | 0.146 | 0.328 | 0.551 | 0.788 |
| $R_\downarrow$ | Baseline | 0.200 | 0.400 | 0.600 | 0.800 |
| | Adaptive | 0.225 | 0.503 | 0.625 | 0.803 |
| $G_\downarrow$ | Baseline | 0.200 | 0.400 | 0.600 | 0.800 |
| | Adaptive | 0.256 | 0.447 | 0.607 | 0.812 |
| $B_\downarrow$ | Baseline | 0.200 | 0.500 | 0.700 | 0.800 |
| | Adaptive | 0.231 | 0.450 | 0.594 | 0.730 |
| $H_\uparrow$ | Baseline | 0.100 | 0.300 | 0.400 | 0.900 |
| | Adaptive | 0.268 | 0.406 | 0.508 | 0.809 |
| $S_\uparrow$ | Baseline | 0.200 | 0.500 | 0.600 | 0.800 |
| | Adaptive | 0.243 | 0.439 | 0.589 | 0.744 |
| $V_\uparrow$ | Baseline | 0.200 | 0.400 | 0.600 | 0.700 |
| | Adaptive | 0.193 | 0.360 | 0.517 | 0.680 |
| $H_\downarrow$ | Baseline | 0.200 | 0.400 | 0.500 | 0.600 |
| | Adaptive | 0.279 | 0.433 | 0.548 | 0.699 |
| $S_\downarrow$ | Baseline | 0.200 | 0.400 | 0.600 | 0.900 |
| | Adaptive | 0.199 | 0.344 | 0.562 | 0.847 |
| $V_\downarrow$ | Baseline | 0.200 | 0.400 | 0.600 | 0.800 |
| | Adaptive | 0.197 | 0.397 | 0.594 | 0.797 |
| $blur$ | Baseline | 9 | 19 | 25 | 35 |
| | Adaptive | 9 | 17 | 23 | 31 |
| $noise$ | Baseline | 10 | 15 | 20 | 30 |
| | Adaptive | 6.4 | 12.4 | 17.7 | 26.9 |

Table 10: **Comparison of computed perturbation levels using a baseline Shen et al. (2021) sensitivity analysis method versus our adaptive method.** $p_5$ is 1 for all RGB/HSV channels, 49 for blur, and 50 for noise. In previous work, each perturbation level is chosen from a certain number of sampled, discretized values. Additionally, these perturbed datasets are generated offline in an additional step before training. Our fast sensitivity analysis enables sensitivity analysis to be performed on the fly during training, and offers much more dynamic, accurate, and descriptive sensitivity curves.

### D.9 KID VS. FID RELATIVE ERROR COMPARISON WITH SCALING SAMPLE SIZES

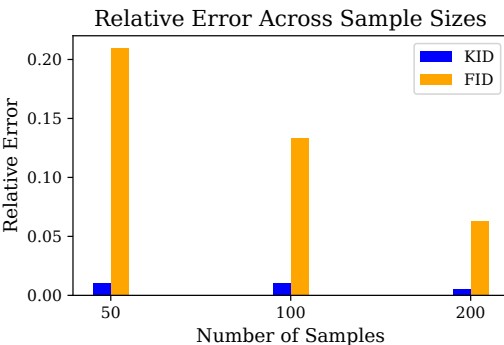

Figure 13: **Relative error of KID and FID over several sample sizes.** We plot the relative error of computed KID and FID values over several sample sizes, with the reference value being the computed value for each at 500 samples. From this, we can see that FID is significantly biased toward the number of samples used for evaluation. We can reduce the evaluation of KID values in sensitivity analysis by a notable fraction due to this property.

### D.10 TRAIN-TIME EVALUATION ON PERTURBED DATASETS

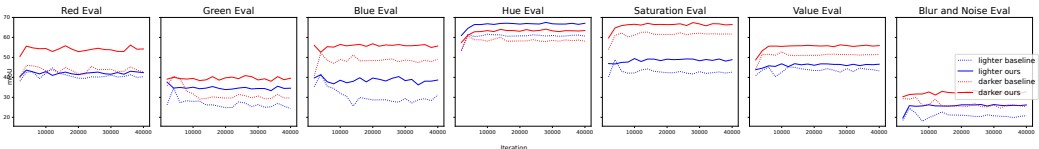

Figure 14: **Evaluation on perturbed test datasets over training iterations.** We show the evaluation on each perturbed dataset during training of our model and the baseline for VOC2012 dataset.

### D.11 ADAPTIVE SENSITIVITY ANALYSIS WITH DIFFERENT NUMBER OF LEVELS

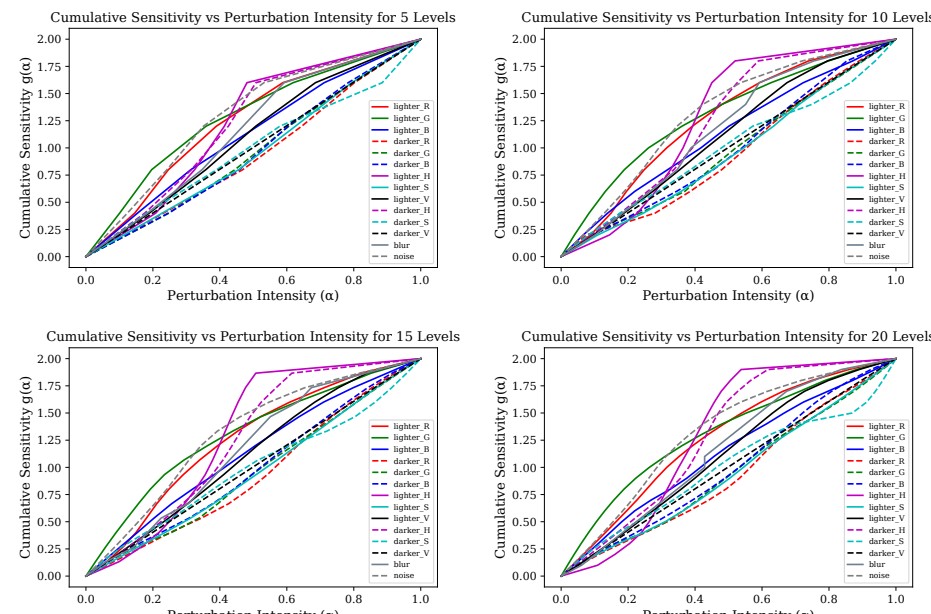

Figure 15: **Visualization of cumulative sensitivity curve with varying number of levels** $L$**.** We visualize the cumulative sensitivity curve in Figure **??** when computing for 5, 10, 15, and 20 levels. We find that even when we increase the number of levels, the curves remain *approximately* the same. Thus, we use 5 levels in our implementation to reduce compute for the sensitivity analysis step.