# OpenReview forum: "Sensitivity-Adaptive Augmentation for Robust Segmentation"
_ICLR.cc/2025/Conference — Submitted to ICLR 2025_

### Official Review · Reviewer_RTLN · 2024-10-27

**Soundness:** 2
**Presentation:** 2
**Contribution:** 3
**Rating:** 3
**Confidence:** 3

**Summary:**

The paper introduces a technique to identify optimal data augmentation parameters that enhance the robustness of a semantic segmentation DNN. The authors focus on augmentations parameterized by a variable, alpha, examining the model's sensitivity to this parameter. To assess sensitivity, they use the Kernel Inception Distance (KID), which varies based on the sample and alpha. Additionally, they define the Metric of Accuracy (MA), which measures performance after image modification, dependent on X and alpha. The objective is to maintain stability in MA while allowing KID to vary. Sensitivity, defined as the ratio of these metrics, is minimized in theory, although, due to computational constraints, they approximate this minimization. Further, to reduce costs, the exploration of alpha is restricted to a truncated parameter space rather than the entire range.

**Strengths:**

The paper addresses a relevant and compelling task of identifying optimal augmentation parameters, and the aim of increasing DNN robustness is beneficial for the field.

**Weaknesses:**

**1. Writing Quality:**

The paper’s writing is difficult to follow and would benefit from thorough revision. I recommend that the authors dedicate more time to clarifying and refining the text.

**2. Missing Comparisons:**

The paper lacks comparisons with relevant methodologies in the field. While the authors include AutoAugment, it is not specifically trained on semantic segmentation, which limits the relevance of the comparison. It would be valuable to benchmark the author’s approach against meta-learning algorithms such as Hyperopt [1], Spearmint [2], Tree-Structured Parzen Estimator (TPE) [6], SMAC [3], Autotune [4], and Vizier [5].
Additionally, comparisons with robust semantic segmentation methods like those in [7, 8, 9, 10] would strengthen the evaluation and provide context for the proposed technique

**3. Mathematical Clarity and Accuracy:**

3.1 Further explanation of the MMD (Maximum Mean Discrepancy) and the choice of kernel, including details on the polynomial kernel, would be helpful.
3.2 Clarification is needed on the transition from Equation 3 to Equation 6.
3.3 Equation 5 appears to have an inconsistency; the denominator linked to KID differs from that in MA—could this be an error?
3.4 The KID numerator in Equation 7 is unusual; please clarify its derivation.
3.5 The definitions of alpha' and alpha'' in Equations 4 and 5 are unclear and require explanation.
3.6 The equation between Equations 8 and 9 seems incorrect. My calculations indicate a result of infinity rather than 2; could the authors verify this?
3.7 Please define T in Equation 10.
3.8 The section from lines 290 to 303 is challenging to interpret even after multiple readings. A rewrite of this section would be beneficial.

**4. Methodology Clarity:**

The paper could improve clarity regarding its approach. The claim of finding data augmentations efficiently is weakened by unclear and potentially biased comparisons. It appears unfair to compare a general approach to an optimized dataset-specific one, and there is also ambiguity in the data augmentations used by each method. Including a table detailing the data augmentations for all algorithms would help in transparency.

**5. Sensitivity Analysis:**

An analysis of the sensitivity of the proposed method to different data augmentation parameters would provide further insight and strengthen the evaluation.


[1] James Bergstra, Brent Komer, Chris Eliasmith, Dan Yamins, and David D Cox. Hyperopt: a Python library for model selection and hyperparameter optimization. Computational Science & Discovery, 8(1):14008, 2015.

[2] Jasper Snoek, Hugo Larochelle, and Ryan P Adams. Practical bayesian optimization of machine learning algorithms. In NIPS, pages 2951–2959, 2012.

[3] Frank Hutter, Holger H. Hoos, and Kevin Leyton-Brown. Sequential model-based optimization for general algorithm configuration. In LION, pages 507–523, 2011. ISBN 978- 3-642-25565-6.

 [4] Patrick Koch, Oleg Golovidov, Steven Gardner, Brett Wujek, Joshua Griffin, and Yan Xu. Autotune: A derivativefree optimization framework for hyperparameter tuning. In KDD, pages 443–452, 2018. ISBN 978-1-4503-5552-0.

 [5] Daniel Golovin, Benjamin Solnik, Subhodeep Moitra, Greg Kochanski, John Karro, and D Sculley. Google Vizier: A service for black-box optimization. In KDD, pages 1487–1495, 2017. ISBN 978-1-4503-4887-4.

 [6] James Bergstra, Remi Bardenet, Yoshua Bengio, and  Balazs Kegl. Algorithms for hyper-parameter optimization.  In NIPS, pages 2546–2554, 2011.

[7] Larsson, M., Stenborg, E., Hammarstrand, L., Pollefeys, M., Sattler, T., & Kahl, F. (2019). A cross-season correspondence dataset for robust semantic segmentation. In Proceedings of the IEEE/CVF Conference on Computer Vision and Pattern Recognition (pp. 9532-9542).

[8] Xu, X., Zhao, H., & Jia, J. (2021). Dynamic divide-and-conquer adversarial training for robust semantic segmentation. In Proceedings of the IEEE/CVF International Conference on Computer Vision (pp. 7486-7495).

[9] Franchi, G., Belkhir, N., Ha, M. L., Hu, Y., Bursuc, A., Blanz, V., & Yao, A. (2021). Robust semantic segmentation with superpixel-mix. arXiv preprint arXiv:2108.00968.

[10] Myronenko, Andriy, and Ali Hatamizadeh. "Robust semantic segmentation of brain tumor regions from 3D MRIs." Brainlesion: Glioma, Multiple Sclerosis, Stroke and Traumatic Brain Injuries: 5th International Workshop, BrainLes 2019, Held in Conjunction with MICCAI 2019, Shenzhen, China, October 17, 2019, Revised Selected Papers, Part II 5. Springer International Publishing, 2020.

**Questions:**

See most of the questions on the weakness.

Also please add a pseudo code describing your technique

---

> ### Author Response · Authors · 2024-11-27
>
> Thank you for your review!  We improved our revisions based on your feedback, and address each point in specific, below.
>
> 1. *Writing Quality.* Thank you for your feedback on writing clarity. We have made substantial revisions to our writing, especially in the methodology section. We hope that the updated text provides more intuition and clarity regarding our approach.
>
> 2. *Missing Comparisons.* Our approach is a data augmentation scheme, similar to the works used in comparisons. Specifically, the AutoAugment used in our comparisons is the same policy trained on ImageNet in the original paper, which requires about 15,000 GPU hours and is shown to transfer strongly to other datasets. In our revisions, we have clarified that the ImageNet policy is used in comparisons and that we do not retrain the augmentation policy.
> While our approach models sensitivity based on the current training data, our augmentation routine is based purely on heuristics and is not a learnable policy. Thus, we cannot “transfer” one augmentation scheme from another dataset to ours, unlike AutoAugment. Our method does not require training an additional model for augmentation selection. Additionally, we evaluate models on corruption benchmarks with scenarios not seen in any model during training. In other words, evaluation benchmarks test generalization capabilities on a third distribution of data, which is neither the source dataset (ImageNet) nor the target dataset (dataset used in training).
> In our original submission, we included a small table of classification result comparisons on the CUB dataset for all methods. This demonstrates that our method still provides improvements over others in classification tasks. The CUB dataset, being much smaller than the ImageNet dataset, would have a much less robust augmentation policy under AutoAugment.
> Lastly, we considered reimplementing AutoAugment. While many public repositories describe existing augmentation policies, none reproduce the reinforcement learning process that trains the policy. As such, we decided to compare the official implementations of AutoAugment.
> Our method indeed has similarities with meta-learning approaches, but we would like to clarify that our approach is a data augmentation scheme explicitly based on “sensitivity modeling” as a function of input data corruption and model output performance. That is, to determine how the amount of variation in input data is related to the amount of variation in model output performance. Meta-learning algorithms describe a class of techniques for generalization across tasks. We emphasize this difference to highlight that our approach can be used orthogonally to meta-learning algorithms, not as a direct comparison.
> We have, however, implemented a more recent work in our revisions: IDBH. IDBH is an augmentation approach published in ICLR 2023 focused on boosting adversarial robustness through data augmentation. This method outperforms previous baselines we compared to in the original submission, and further strengthens our improvements.
>
> 3. *Mathematical and Methodology Clarity.* This feedback was helpful, and we have clarified these points in a simplified and rewritten methodology section. We emphasize that the original approach remains the same, but notation and explanations have been reworded for clarity to address reviewer’s suggestions.
>
> 4. *Sensitivity Analysis.* Per this feedback, similarly to that of other reviewers, we have included sensitivity analysis plots across training of Cityscapes, showing how sensitivity changes with respect to RGB, HSV, blur, and noise channels. Since the ablation study shows that photometric augmentations contribute most to generalization, this additional analysis (Figure 4) provides insight into which color channels contribute most to generalization. In the case of Cityscapes, we note that the Hue channel plays a large role in generalization, due to its volatility during training as the model generalizes. Exploring this avenue further would be interesting and beneficial for model interpretability.

---

> > ### Comment · Reviewer_RTLN · 2024-11-30
> >
> > I have taken some time to revisit the paper, and I appreciate the effort in trying to address my previous questions. However, I must say that the paper remains unclear to me. The positioning and contribution of the work are not well-defined and the missing comparisons does not convince me.
> >
> > I strongly recommend that the authors carefully revise the paper, clearly emphasizing their contribution and providing an experimental protocol that effectively evaluates it. As it stands, the paper seems to sit somewhere between data augmentation for robust semantic segmentation and sensitivity analysis, without clearly committing to a specific focus.
> >
> > I would prefer the paper to concentrate on a single topic and explore it thoroughly, ensuring all comparisons are fairly addressed within that context. For instance, if the focus is robustness, it’s crucial to include fair comparisons with other works in the robustness domain. Conversely, if the focus is sensitivity analysis, robustness comparisons are unnecessary. The current approach creates confusion.
> >
> > I suggest the authors take the time to rewrite the paper—or perhaps consider if the work would be better split into two distinct papers. This raises the question: are there two separate contributions here, or just one?

---

> ### Author Response · Authors · 2024-11-30
> **Contributions**
>
> Dear Reviewer RTLN:
> *Similar to many, many published papers*, this work offers multiple key contributions:
> 1. An efficient *adaptive sensitivity analysis* method for online model evaluation that iteratively approximates model sensitivity curves for speedup;
> 2. A comprehensive framework that leverages sensitivity analysis results to systematically improve the robustness of learning-based segmentation models;
> 3. Evaluation and analysis of our method on unseen synthetically perturbed samples, naturallycorrupted samples, and ablated contributing factors to robustification.
> Our newly derived theoretical framework on adaptive sensitivity analysis runs up to *10×* faster and requires up to *200×* less storage than previous approaches, enabling practical, on-the-fly estimation during training for a model-free augmentation policy.  But, it's too *mathy* for most readers, in spite of sensitivity analysis is a very intuitive concept (how the input varies would impact the output of learning or black-box system).   So demonstration on at least 1 application is expected by reviewers.
>
> You're right that the contributions here are probably worthy of 2 papers and not incremental at all.   The first contribution has tremendous impact on many other areas of learning-based visual processing, control, planning, synthesis and much more, but it's not possible to demonstrate this novel concept of on-the-fly sensitivity analysis without demonstration on at least ONE application and/or task.  Just 1 application to segmentations requires so many comparisons and multiple man-years and the results in Appendices are already longer than the paper itself.  We would appreciate your reconsideration of the potential impact of such work for the broader learning community.   All learning algorithms can benefit from faster and leaner training.  We will release the code upon acceptance of this work, so others will enjoy the same benefits of (nearly) free data-augmentation for robustification of learning-based algorithms beyond segmentation -- though immediately the results on segmentation already significantly advance the state-of-art.
>
> To the concern: "the paper seems to sit somewhere between data augmentation for robust semantic segmentation and sensitivity analysis, without clearly committing to a specific focus", we want to highlight that our work makes online sensitivity analysis fast and viable for use in data augmentation schemes. Data augmentation is one of many ways to increase generalization capabilities of a neural network. While other works approach robustness from adversarial objectives or from the architecture side, our work focuses on the data side. Other methods for improving robustness, such as the adversarial techniques cited, can be implemented orthogonally to our method in the same framework. For our augmentation scheme, which is informed by a sensitivity heuristic, we choose to compare to other data augmentation approaches which boost generalization.
>
> Our work has many contributions, but we believe that all contributions are related for one unified work.
>
> Thank you!
>
> Sincerely,
>
> The Authors

---

### Official Review · Reviewer_yZkL · 2024-11-03

**Soundness:** 3
**Presentation:** 2
**Contribution:** 3
**Rating:** 6
**Confidence:** 4

**Summary:**

The paper presents a approach for improving the robustness of segmentation models through sensitivity-based data augmentation. It attempts to address the challenges of natural corruptions in image segmentation by combining sensitivity analysis with data augmentation. The use of Kernel Inception Distance (KID) to measure the extent of augmentation and the formulation of the sensitivity analysis objective are well-motivated and provide a new perspective on improving model robustness. Expeirmental results confirm that the proposed augmentation method improves model's robustness in real-world and synthetic datasets.

**Strengths:**

1. The idea of using sensitivity analysis to adaptively augment sample is interesting and has the potential to enhance the performance of segmentation models in the presence of natural corruptions.

2. The authors conduct a series of experiments on multiple datasets and in different scenarios, including real-world corruptions (ACDC and IDD datasets), various synthetic benchmarks (ADE20K, VOC2012, etc.), and fine-tuning with foundation models in downstream tasks. The comparison with several state-of-the-art augmentation methods helps to demonstrate the effectiveness of the proposed method.

3. The paper acknowledges the limitations of the current work, such as the inability to handle physical occlusions and the relatively poor performance in low-lighting scenarios. This shows the authors' awareness of the potential areas for improvement and provides a direction for future research.

**Weaknesses:**

1. The description of sensitivity analysis, especially the calculation of the sensitivity curve and the interpretation of its derivative, could be more intuitive. The equations and concepts related to sensitivity analysis may be difficult for some readers to understand without further clarification. For example, the relationship between the MA-KID curve and the model's sensitivity is not entirely clear from the current explanation.

2. The proposed fast sensitivity analysis is the key component of the paper, but its description is rather technical and lacks an intuitive explanation of why it works and how it significantly reduces the computational cost.

3. The iterative estimation process and the use of Piecewise Cubic Hermite Interpolating Polynomial (PCHIP) could be better illustrated with examples or visualizations.

4. In comparison with other data augmentation techniques, the paper could explore more about the reasons why some methods perform better or worse in different scenarios. For instance, a deeper analysis of why TrivialAugment has certain limitations compared to the proposed method could provide more insights into the novelty and superiority of the proposed approach.

5. Fig. 1 is confusion. It looks more like a result of the analysis and does not present the complete pipeline of the method. I suggest redrawing Fig. 1 so that this figure can reflect the steps of the method (Overview of the proposed method).

**Questions:**

See "Weaknesses"

---

> ### Author Response · Authors · 2024-11-27
>
> Thank you for your feedback! From the comments, we understand that a major issue was the clarity of presentation and lack of analyses. We are happy to update the reviewer with new paper revisions incorporating their feedback. Specific points are addressed below.
>
> - *The description of sensitivity analysis could be more intuitive.* Based on this feedback, we have updated the explanation of our methodology, along with simplified notation and intuitive explanation. Algorithm pseudocode for training has been added to the main text (Algorithm 1).
>
> - *The proposed fast sensitivity analysis lacks an intuitive explanation of why it works and how it significantly reduces the computational cost.* Similar to the point above, we have revised our methodology section to address these concerns. For convenience, we reiterate here that the max-min objective prioritizes augmentation intensity values to be densely sampled in regions of high sensitivity (i.e., high change in accuracy, and low changes in KID). The computational cost reduction stems from an additional outer loop in previous work, which discretized the augmentation space uniformly along the augmentation value space, then optimized amongst the discretized values. Due to this offline sampling step, the storage necessity and computational complexity is much greater.
> Our adaptive sensitivity analysis focuses data augmentation near regions of high sensitivity, i.e. small variation in input would lead to high variation in the output performance.  So, by sampling more around high-sensitivity regions and much less in low-sensitivity regions can lead to significant cost saving in the amount of augmented data, thus significantly reducing the computation and the storage.
>
>
> - *The iterative estimation process could be better illustrated with examples or visualizations.* Our original submission contained a figure demonstrating this process, but we neglected to reference it in the main body. In our revisions, we have referenced Figure 12 in Section D.7 of the appendix, which describes the iterative estimation process involving PCHIP.
>
> - *The paper could explore more about the reasons why some methods perform better or worse in different scenarios.* TrivialAugment is a simplistic augmentation approach that can be interpreted as uniformly and randomly sampling augmentation intensities. In our updated revisions, we included ablation study results showing several variations of our method and the contributions of each component. For our method, we believe that the largest boost in performance comes from sensitivity analysis itself; without it, the random augmentation scheme is equivalent to TrivialAugment. Additionally, we moved analyses from the appendix to the main body to highlight insights into how the sensitivity of each color channel changes as generalization improves during training. From the results, we find that Hue channel plays a significant role in generalization capabilities in the case of Cityscapes.
>
>
> - *Fig. 1 is confusing.* Per this feedback, we have re-drawn Figure 1 to include all components of our method in more detail. We hope that this revision provides more clarity to the reviewer!

---

> > ### Comment · Reviewer_yZkL · 2024-11-27
> > **Post-Rebuttal**
> >
> > Thanks for the response from the authors. I have carefully checked the response from the authors, and most of my concerns have been addressed. I am happy to maintain my initial score.

---

> > > ### Author Response · Authors · 2024-11-27
> > >
> > > We are glad to hear that most of your concerns were addressed! In addition to your feedback, reviewers had brought up concerns with recent comparisons and lack of ablation studies as well. Thus, our revised draft has included major changes (highlighted in red) compared to the initial submission. Are there any additional improvements we can make to the revised version?

---

### Official Review · Reviewer_2HNr · 2024-11-04

**Soundness:** 3
**Presentation:** 2
**Contribution:** 3
**Rating:** 5
**Confidence:** 4

**Summary:**

The manuscript proposes a novel sensitivity-based data augmentation method to enhance the robustness of semantic segmentation models against natural corruptions. This method significantly improves upon existing approaches in terms of speed and storage efficiency, with runtimes up to 10 times faster and storage requirements reduced by 200 times. It dynamically adjusts augmentations based on the model's sensitivity, thereby enhancing robustness across various real-world and synthetic datasets.

**Strengths:**

1. The paper proposes an efficient sensitivity analysis method, which is 10 times faster and reduces storage requirements by 200 times compared to existing methods. This improvement is particularly useful during online training.
2. Through gradient-free sensitivity analysis and dynamic sampling techniques, this method enables real-time data augmentation during training without the need for additional models or complex computational resources.

**Weaknesses:**

1. Lack of specific ablation experiments; the performance improvement could be attributed to the pre-trained weights rather than the method proposed in the manuscript. The absence of detailed ablation comparisons may lead to confusion.
2. The comparison is limited to only a few existing data augmentation methods, lacking a comparison with the latest methods from 2024 to comprehensively assess its superiority.
3. As the authors mentioned, they did not fully consider the performance of the method under low-light conditions. However, in complex weather like snow, low-light conditions often accompany, making it challenging to assess its contribution to the community.
4. The paper treats all types of augmentations as equivalent, but is the contribution of all data augmentations the same in practical environmental conditions? The authors might need to attempt visualizations to specifically present this.
5. The explanation of hyperparameters is not detailed, including but not limited to the settings of hyperparameters on the original model (although this could be elaborated in the appendix, it seems the authors overlooked this). Also, are the same hyperparameters used across different datasets and environmental conditions?
6. There are some minor writing issues, such as most formulas lacking punctuation, and it is suggested that the last page of the manuscript should not be left blank.

**Questions:**

1. Can the method maintain robustness when dealing with other types of disturbances beyond natural ones, such as artificial image editing?
2. What are the computational resource requirements for the method mentioned in the manuscript on datasets of different scales? Is it possible to effectively deploy it in resource-constrained environments?
3. What challenges does the proposed method face in handling physical occlusions (such as windshield wipers)? Can physical occlusions be simultaneously treated as a complex environment to address?
4. What are the computational resource requirements for the method mentioned in the paper on datasets of different scales? Is it possible to effectively deploy it in resource-constrained driving environments?

---

> ### Author Response · Authors · 2024-11-27
>
> Thank you for your insight! We are happy to share that we incorporated most of the weakness feedback into our revisions.
>
> - *Lack of specific ablation experiments.* We have added an ablation study in Section 4.5. Regarding pre-trained weights, all experiments are initialized with the same SegFormer weights pre-trained on ImageNet-1k, including baseline experiments. Thus, all performance improvements are fair with respect to model initializations. We also fix the random seed for all experiments to ensure reproducible results across runs.
> - *The comparison is limited.* To improve on this feedback, we added an additional comparison to IDBH, proposed by Li et al., and published in ICLR 2023: https://openreview.net/forum?id=y4uc4NtTWaq. We have updated our results tables to include this comparison and maintain that our method produces better results in unseen corruption scenarios, both artificial and real-world.
> - *Consideration of low-light settings.* We would like to clarify that our experiments evaluate models trained on clear, sunny weather on datasets of adverse weather, including that of nighttime data. Since nighttime data is not involved in training, we expect performance to be lower in these conditions. While augmentation for all comparisons include brightness augmentations, this alone is not sufficient to address the performance gap in night time data. We note this limitation of our work and consider it a promising area for future research, along with exploration of photometric perturbations and their impact on model generalization.
> - *The paper treats all types of augmentations as equivalent.* To address this, we added an experiment in our ablation study to examine whether removing the assumption of equivalence among augmentations affects performance. While the performance is largely the same, we do observe slight improvements. We thank the reviewer for pointing out this assumption and have addressed it in the revision. For visualizations, we moved Figure 4 from the appendix of the original submission to the main body to illustrate how different color channel sensitivities change over training. Interestingly, we note that Hue may play a significant role in model generalization for segmentation.
> - *The explanation of hyperparameters is not detailed.* We take reproducibility seriously and thank the reviewer for pointing this out. We have added a hyperparameter table (Table 7) to the appendix, detailing differences between augmentation techniques and training parameters for experiments. We also clarified in the main body that training parameters for all methods are the same, with the main differences being the data augmentation schemes. The set of augmentation functions across all methods is the same, with exception to IDBH, which involves two extra augmentations (RandomFlip, RandomErasing). We maintain these two additional augmentations to stay true to the authors’ original implementation.
> - *Minor writing issues.* We have revised writing issues throughout the paper and hope that the improved clarity is reflected in the updated version!

---

> > ### Author Response · Authors · 2024-11-27
> >
> > Regarding the questions as well:
> >
> > - *Can the method maintain robustness when dealing with other types of disturbances beyond natural ones, such as artificial image editing?*
> > Yes, our method maintains robustness even in artificial disturbances. One specific benchmark of synthetic scenarios in our results is the ImageNet-C (IN-C in tables). For more sophisticated image editing benchmarks, the most recent and systematic benchmark is from CVPReviewer 2HNr024: https://github.com/PRIS-CV/Pascal-EA. However, this code was released 2 months ago, after the original paper submission deadline, with no released dataset. Given more time, we can generate evaluation data per their method and include attribute editing results in future revisions.
> >
> > - *What are the computational resource requirements for the method mentioned in the manuscript on datasets of different scales? Is it possible to effectively deploy it in resource-constrained environments?* Yes! In fact, our method is designed to be lightweight and portable for data augmentation. All experiments in the paper are trained on 4 RTX A4000 GPUs in parallel. The only additional overhead is during sensitivity analysis updates, which takes about 2.8 times the duration of a regular evaluation pass. In our experiments, sensitivity analysis updates occur about 3 times during training for an experiment spanning 160k training iterations. Otherwise, augmentations are applied similarly to any other method, with little overhead to vanilla training. We show data efficiency results in Figure 3 to demonstrate data-constrained settings.
> >
> > - *What challenges does the proposed method face in handling physical occlusions (such as windshield wipers)? Can physical occlusions be simultaneously treated as a complex environment to address?* Occlusions and random erasing are not included during training for our method but can be easily included on top of our current approach. To address this question, we have added qualitative visualizations of windshield wipers in rainy weather to the main body of the paper (Figure 2). With the updated SegFormer backbone in revision results, we show qualitatively that our method handles windshield wipers well compared to a recent method (IDBH) that explicitly includes random erasing in the training process.

---

> > > ### Comment · Reviewer_2HNr · 2024-11-30
> > >
> > > Thank you for your detailed answers, which addressed some of my concerns. Considering the other reviewers' discussion, I maintain my original rating.

---

> > > > ### Author Response · Authors · 2024-11-30
> > > >
> > > > Thanks for the response! Could you please provide us more detail as to which concerns were left unclarified? Perhaps we can discuss them. Also, just a friendly reminder that we have uploaded a new revision which incorporates all reviewers’ feedback.

---

### Official Review · Reviewer_yBnW · 2024-11-04

**Soundness:** 3
**Presentation:** 3
**Contribution:** 2
**Rating:** 5
**Confidence:** 5

**Summary:**

This work aims to optimize data augmentation policies to enhance semantic segmentation, particularly targeting domain-shift issues. The proposed approach utilizes a derivative-based local method (first-order indices), approximated with Monte Carlo, to derive importance weights for each augmentation policy. Subsequently, all policies are sampled based on their importance and ensembled to create an augmented data pool. The resulting performance gains are validated through experiments across various domain-shift datasets, demonstrating the effectiveness of the method.

**Strengths:**

1. The experiments demonstrate that augmentation with policy optimization effectively enhances semantic segmentation modeling.
2. The proposed approach is presented in a clear and organized manner, making it easy to follow. The appendix provides valuable details on the augmentation configuration, enhancing understanding.

**Weaknesses:**

1. The contributions and insights of this work are somewhat limited. It primarily focuses on enhancing performance through augmentation policy optimization. The sensitivity assessment resembles model uncertainty computation, and uncertainty-weighted segmentation modeling has already been developed to improve model performance. Additionally, this work may raise concerns about relying on performance-boosting tricks rather than providing critical scientific research insights.
2. The experiments lack alignment in terms of using the same semantic segmentation models. The varying choices, without clear justification, could raise concerns about the potential cherry-picking of results. The chosen baseline models, particularly for clean image evaluation, are relatively weak and may not adequately reflect state-of-the-art performance.
3. The augmentation benchmarking experiments are not particularly convincing. The chosen baselines focus on the design of augmentation algorithms, while the proposed method resembles an ensemble of multiple augmentation algorithms, optimizing policy ensembling with importance weighting. Also, the policy optimization is quite common in machine learning model training. It would be beneficial to compare this approach with other policy-based optimization methods.

**Questions:**

The weaknesses mentioned above raise several questions. The rebuttal should address these concerns to improve the rating.

---

> ### Author Response · Authors · 2024-11-27
>
> Thank you for your feedback! We included revisions to our submission based on your suggestions.
>
> - *The contributions and insights of this work are somewhat limited.* While we disagree with the assertion that our method relies on performance-boosting tricks, we acknowledge that we may have presumed ‘sensitivity analysis’ to be a well-known concept, which may have led to a lack of intuitive explanation and detailed insights in our approach. In fact, incorporating sensitivity analysis as an explicit step in data augmentation can offer more transparency to model robustness throughout training. To address weakness point #1, we added Figure 4 in our results to show how sensitivity analysis can be useful to draw insights on model generalization.
>
> - *The experiments lack alignment in terms of using the same semantic segmentation models.* We have replaced all experiments using PSPNet in the main body with those using SegFormer, which is a recent state-of-the-art architecture for robust segmentation. This ensures consistency of architecture across experiments. Our original intention in showing results on both architectures was to demonstrate that our method produces performance boosts on different backbones—however, we understand that this may have caused more confusion than clarity. Results comparing PSPNet and SegFormer performance using our method can still be found in Section D.5 of the appendix.
>
> - *The augmentation benchmarking experiments are not particularly convincing.* We would like to clarify a misunderstanding. Our method is not an ensembling technique of multiple augmentation algorithms; rather, it is a single routine that models sensitivity for several different augmentation types. It is as much as an ensemble as random augmentation (TrivialAug) would be considered an ensemble of uniform distributions. We also want to note that while our method determines an augmentation scheme which prioritizes sensitive augmentations, it differs from common policy optimization approaches in that it is neural-network free. Our approach is purely based on sensitivity-guided performance optimization and only the segmentation model itself is being optimized.

---

### Author Response · Authors · 2024-11-27
**Thank you to all reviewers! We have updated our submission based on your feedback.**

We thank the reviewers for their time and actionable feedback. We have made considerable efforts to incorporate these requests into the paper revisions! We appreciate the reviewers’ patience and time, which have been highly valuable to improving the exposition of this work.

We are pleased that reviewers found “sampling-based sensitivity analysis” to be interesting (Reviewer yZkL), the motivation to be compelling (Reviewer RTLN), organization and appendix to be helpful (Reviewer yBnW), and acknowledged on efficiency savings (Reviewer 2HNr). We have thoroughly discussed all feedback among the authors and have addressed it in our revisions. Below, we address concerns raised by multiple reviewers.

One common concern was the clarity of presentation. We agree that a major source of confusion stems from a lack of an intuitive explanation of sensitivity analysis and how it benefits robustness, which hindered the interpretation of our positive results. To amend this, we have revised our methodology section in three main areas: 1) improved notation (Reviewer 2HNr, Reviewer RTLN) and provided more intuitive explanations in the text (Reviewer yZkL); 2) moved the pseudocode from the appendix to the main body text (Reviewer RTLN); and 3) added a diagram illustrating the iterative process involving PCHIP (Reviewer yZkL).

Another general concern was the lack of insight to the scientific takeaways of sensitivity analysis—for example, why some methods perform better or worse in different scenarios (Reviewer yZkL), and the need for  ablation studies to explain what contributes to performance gain (Reviewer 2HNr). We believe that this feedback is directly related to  Reviewer yBnW’s concern about relying on performance-boosting tricks rather than providing critical scientific research insights. In our revision, we have included additional quantitative results evaluating the sensitivity of other augmentation approaches, such as TrivialAugment (Reviewer yZkL), and have added ablation study results to show how each component influences performance (Reviewer 2HNr).

A major concern from reviewers was on the transparency of experimental parameters (Reviewer yBnW, Reviewer 2HNr, Reviewer RTLN)—we take this concern seriously with respect to both fair comparison and reproducibility. We had stated in the methodology section that all experiments are run in similar settings, with the same hardware. However, we agree that this is not detailed enough, especially for reproduction of results, leading to concerns of unfair comparisons with reviewers (Reviewer yBnW, Reviewer RTLN). To address this, we have added a table of hyperparameters for each experiment in the appendix, and referenced it in the main body text for revisions. Additionally, we plan to release implementation code and all corresponding experiment configurations upon acceptance of this work.

Lastly, reviewers also mentioned a lack of comparison to more recent methods (Reviewer 2HNr, Reviewer RTLN). In our experiments, we benchmarked our method against other popular augmentation methods, all of which have varying augmentation policies (some random, and some driven by a NN-based policy). We agree that additional results on recent augmentation methods can further strengthen our claims on performance superiority. As such, we have included results on IDBH (https://openreview.net/forum?id=y4uc4NtTWaq), an augmentation approach for adversarial robustness published in ICLR 2023, which can be considered for direct comparison for our method.

Please let us know if our revisions have addressed your concerns. Thank you, again!

---

### Meta-Review · Area_Chair_o9Tu · 2024-12-17

**Metareview:**

The paper introduces a sensitivity-based augmentation method to improve segmentation robustness against corruptions. The strengths: interesting sensitivity-driven augmentation, high computational efficiency, and performance improvements. However, some critical issues are pointed out in the reviews: limited comparisons with recent augmentation and robustness methods, method lacks clarity, limited ablation studies and weak experiments. While the proposed method shows high efficiency, weaknesses in clarity, limited comparisons, and incomplete analyses hinder its overall impact. The paper shows promise but requires stronger baselines, more convince experiments, and clearer presentation. For these reasons, the recommendation is reject. The authors are encouraged to consider the reviewers' comments when revising the paper for submission elsewhere.

**Additional Comments On Reviewer Discussion:**

The reviewers keep the rating on discussion. While the authors made improvements and addressed several critical points, unresolved concerns about clarity, comparisons, and focus weigh against the paper. These limitations reduce its overall impact. As a result, the final recommendation is reject.

---

### Decision · Program_Chairs · 2025-01-22

Reject